# Ion mobility-based sterolomics reveals spatially and temporally distinctive sterol lipids in the mouse brain

Tongzhou Li[1,2], Yandong Yin[1], Zhiwei Zhou [1,2], Jiaqian Qiu[1,2], Wenbin Liu[1], Xueting Zhang[1,2], Kaiwen He[1], Yuping Cai[1] & Zheng-Jiang Zhu [1✉]

Aberrant sterol lipid metabolism is associated with physiological dysfunctions in the aging brain and aging-dependent disorders such as neurodegenerative diseases. There is an unmet demand to comprehensively profile sterol lipids spatially and temporally in different brain regions during aging. Here, we develop an ion mobility-mass spectrometry based four-dimensional sterolomics technology leveraged by a machine learning-empowered high-coverage library (>2000 sterol lipids) for accurate identification. We apply this four-dimensional technology to profile the spatially resolved landscapes of sterol lipids in ten functional regions of the mouse brain, and quantitatively uncover ~200 sterol lipids uniquely distributed in specific regions with concentrations spanning up to 8 orders of magnitude. Further spatial analysis pinpoints age-associated differences in region-specific sterol lipid metabolism, revealing changes in the numbers of altered sterol lipids, concentration variations, and age-dependent coregulation networks. These findings will contribute to our understanding of abnormal sterol lipid metabolism and its role in brain diseases.

---

[1] Interdisciplinary Research Center on Biology and Chemistry, Shanghai Institute of Organic Chemistry, Chinese Academy of Sciences, Shanghai, China. [2] University of Chinese Academy of Sciences, Beijing, China. ✉email: jiangzhu@sioc.ac.cn

Sterol lipids represent an important class of lipids and play critical roles in many physiological processes such as cell signaling and energy homeostasis[1–4]. The metabolic homeostasis of sterol lipids is vital to maintain proper cellular and systematic functions of living organisms[3]. Dysregulation of sterol lipid metabolism has been implicated in many human diseases such as cardiovascular diseases[3,5], cancers[4], and neurodegenerative diseases[6]. In mammals, the brain is enriched with a diverse range of sterol lipids. For example, the human brain contains ~20% whole-body cholesterol[7]. In the brain, cholesterol can be synthesized only in situ and serves as the metabolic precursor for conversion to other important sterol lipids such as oxysterols, steroid hormones, and vitamin D[2,8,9]. These sterol lipids are biologically active and crucial for maintaining proper brain functions. Mounting evidence shows that aberrant sterol lipid metabolism in the brain is tightly linked to brain aging[10] and aging-dependent neurodegenerative diseases, such as Alzheimer's disease (AD)[11–13], Parkinson's disease (PD)[14], and Huntington's disease (HD)[6]. Within the brain, sterol lipids are unevenly distributed across different functional regions[15,16]. The changes in sterol lipids associated with brain disorders have also been shown in brain region-specific pathologies[17,18]. For example, the levels of 27-hydroxycholesterol (27-HC) and sitosterol are increased in brain regions frontal cortex and basal ganglia in patients with AD, but no differences are observed in the brain region pons[11]. Importantly, age is a considerable risk factor for the development of neurodegenerative brain disorders, and abnormalities in sterol lipid metabolism are closely coupled to age-related brain dysfunction[19]. Therefore, in-depth characterization of spatial and temporal sterol profiles across the complex functional regions of the brain, which has not yet been examined, would provide a more holistic understanding of the pathologies of brain diseases underpinned by sterol lipid metabolism.

The analysis of sterol lipids in biological samples using gas chromatography-mass spectrometry (GC-MS)[20] and liquid chromatography-mass spectrometry (LC-MS)[21–23] technologies is hindered by multiple factors, such as low concentrations, poor ionization efficiency, and limited isomer separation. Chemical derivatization is commonly employed to increase ionization efficiency and sensitivity, especially for LC-MS-based sterol analysis. Derivatization reagents such as picolinic acid (PA)[21,23], Girard's reagent P (GP)[15], and others[24,25] were systematically developed. However, the enormous structural similarity of sterol lipids still presents a significant challenge for their analysis on a large scale. The statistical analyses have shown that as many as 86% of sterol lipids in the LIPID MAPS Structure Database (LMSD) have isomers (Fig. 1a). Recently, ion mobility-mass spectrometry (IM-MS) has emerged as a promising technique for metabolomics[26–28] and lipidomics[29–31] by providing multi-dimensional separation and high selectivity. Importantly, ion mobility can rapidly separate ions based on differences in their rotationally averaged surface area or collision cross-section (CCS), which is related to the structure and conformation of ions. It contributes to the separation of isomeric compounds that commonly exist in biological samples. Coupling IM-MS with LC and tandem MS technology (e.g., LC-IM-MS/MS) enables the comprehensive and simultaneous acquisition of information on small molecules in biological samples in four dimensions, including the $m/z$ of MS1, retention time (RT), CCS, and MS/MS spectra. For sterol analysis, previous reports have demonstrated the application of IM-MS to successfully distinguish isomers, such as testosterone and dehydroepiandrosterone[32], and 25-hydroxyvitamin D2 and D3[33]. In addition, our group and others developed various CCS databases such as AllCCS[34], ISiCLE[35], and others[36] to support the IM-MS-based analyses of endogenous small molecules including sterol lipids. However, the in-depth characterization of sterol lipids on a large scale with IM-MS has not been achieved due to the lack of a comprehensive four-dimensional database for sterol lipids. To advance the understanding of biological activities implicated by age-associated sterol metabolism in the brain, it is imperative to develop a strategy to support the comprehensive profiling of sterol lipids with high accuracy and broad coverage.

In this study, we first develop an IM-MS-based four-dimensional sterolomics technology leveraged by a machine learning-empowered library for the comprehensive analysis of sterol lipids. We use picolinic acid derivatization to enhance the sensitivity and improve the separation of sterol isomers (Fig. 1b). Then, we develop an LC-IM-MS/MS method for the comprehensive four-dimensional characterization of sterol lipids in complex biological samples (Fig. 1c). Sterols are identified with high accuracy and broad coverage by matching with our four-dimensional sterol library (Fig. 1d). We utilize this four-dimensional technology to profile the spatially resolved landscape of sterol lipids in ten functional regions of the mouse brain. Finally, we extend the spatial sterol lipid analysis to the context of mouse brain aging and delineate brain region-specific and age-associated changes in sterol metabolism. These results reveal that the diverse functions of brain regions require different regulation of sterol metabolism and provide a holistic understanding of spatial and temporal sterol lipid metabolism.

## Results

**Improved separation of sterol isomers with derivatization and IM-MS.** Most sterol lipids have low concentrations in biological samples, and chemical derivatization is commonly employed to increase sensitivity. Since most sterol lipids have a hydroxyl group (88%), picolinyl derivatization was performed (Fig. 1a, b). In addition to enhanced sensitivity, picolinyl derivatization significantly improved the ion mobility separation of sterol isomers. Taking the isomer pair of epiandrosterone and etiocholanone as an example, the CCS difference between the two isomers was enlarged from 0.4% to 12.8% after derivatization (Fig. 2a). We further investigated this effect using a sterol library with 97 sterol lipids. In total, 163 and 125 pairs of sterol isomers were assigned to underivatized and derivatized sterols, respectively (see "Methods" section). The statistical analysis showed that, after derivatization, 67.2% of the sterol isomers had significant CCS differences (i.e., ≥2%). As a comparison, only 38.0% of underivatized sterol isomers have significant CCS differences (Fig. 2b and Supplementary Data 1). It is worth noting that for underivatized sterol lipids of which the major adduct is [M + H–$H_2O$]$^+$ or [M + H–$2H_2O$]$^+$, the loss of water molecules may cause loss of isomerism. More examples are provided in Supplementary Fig. 1a.

Then, we demonstrated that two-dimensional separation based on LC-IM could further improve the separation of derivatized sterol isomers. Specifically, we calculated the peak resolutions ($R_S$) of sterol isomers separated by IM, LC, and the LC-IM-based two-dimensional method[37,38] (Supplementary Data 2). For example, the isomers of aldosterone and cortisone had $R_S$ values of 0.9 and 0.8 obtained in the IM and LC separation groups, respectively. The peak resolution increased to 1.2 with the LC-IM-based two-dimensional separation (Fig. 2c). We also investigated this effect using our in-house sterol library. For the 125 pairs of derivatized sterol isomers, ~80.8% of the isomer pairs achieved good peak resolution ($R_S$ ≥ 1.0; referring to a 98% baseline separation) by LC-IM-based two-dimensional separation. However, only 42.4% and 68.8% of the isomer pairs obtained good peak resolution with IM separation and LC separation alone, respectively (Fig. 2d). More examples are provided in Supplementary Fig. 1b. Together,

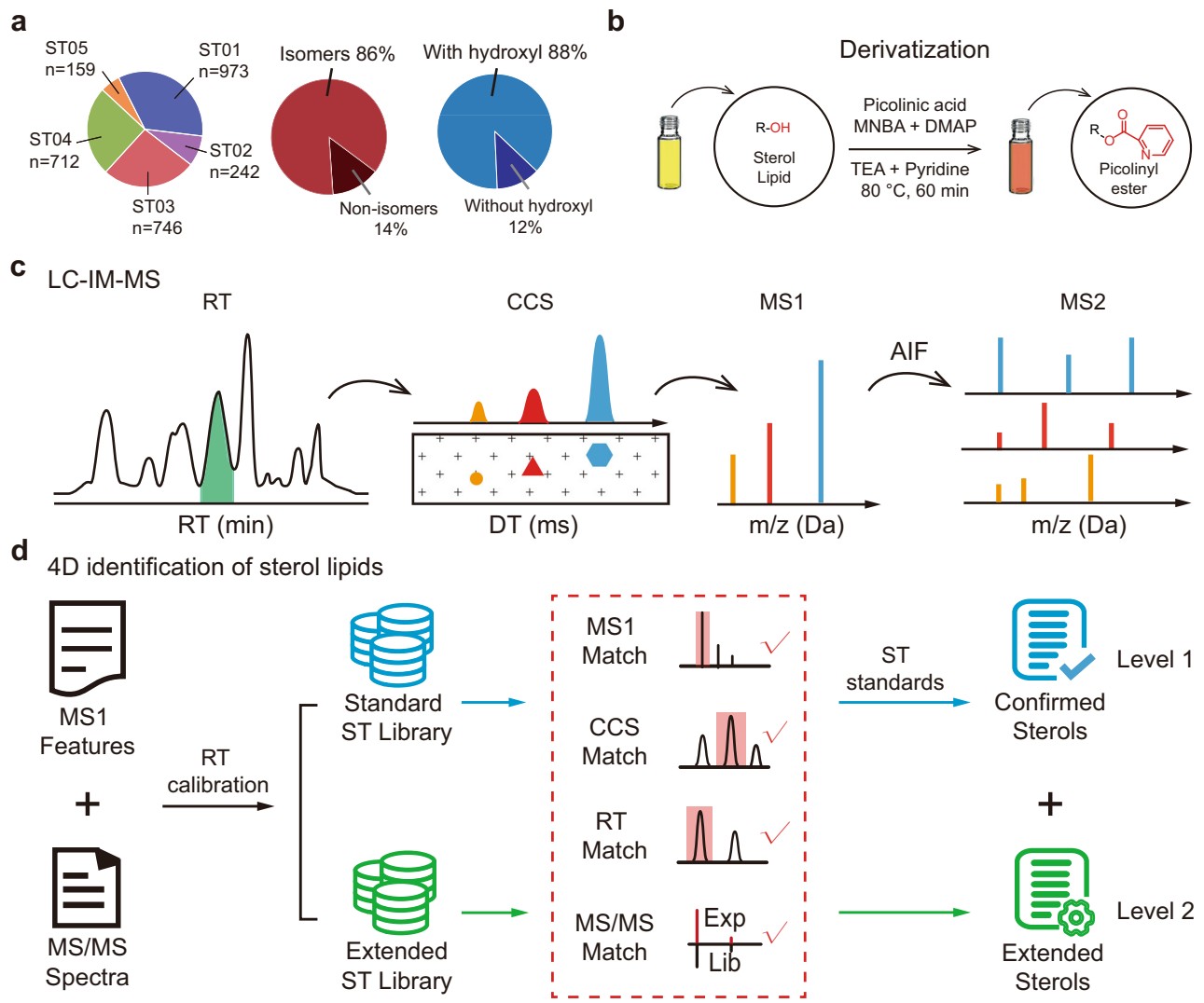

**Fig. 1 Ion mobility-mass spectrometry technology for the four-dimensional analysis of sterol lipids. a** Statistical analysis of sterol lipids in the LIPID MAPS Structure Database (*n* = 2832). Left panel, numbers of sterol lipids in four main classes: ST01, sterols; ST02, steroids; ST03, secosteroids; ST04, bile acids, and derivatives; ST05, steroid conjugates. Middle panel, percentages of sterol isomers in the database. Right panel, percentages of sterol lipids with a hydroxyl group in the database. **b** Schematic illustration of picolinic acid derivatization. **c** Four-dimensional data acquisition enabled by liquid chromatography-ion mobility-mass spectrometry (LC-IM-MS). **d** Sterol identification by matching with the four-dimensional sterol library.

our results demonstrated that picolinyl derivatization improved the separation of sterol isomers for sterol lipid analysis by leveraging multidimensional separation with LC-IM-MS technology.

**Machine learning-enabled four-dimensional library for sterol lipid analysis.** LC-IM-MS/MS technology allows for the comprehensive four-dimensional characterization of sterol lipids in complex biological samples. A four-dimensional sterol library is a prerequisite to support accurate sterol identification on a large scale. First, we collected precursor *m/z* values, RTs, CCS values, and MS/MS spectra from the picolinyl-derivatized standards of 97 sterol lipids and constructed the standard ST library (Fig. 3a and Supplementary Data 3). Then, we curated an extended four-dimensional ST library using a machine learning-based strategy reported in our four-dimensional lipidomics studies[39–41] (Fig. 3a and Supplementary Data 3). Dimension 1 was the MS1 library with 2068 sterol lipids retrieved from the LIPID MAPS Structure Database (LMSD). Dimensions 2 and 3 were the predicted RT and CCS libraries, respectively, which were curated using the

support vector regression (SVR)-based machine learning algorithm (see "Methods" section). In brief, 57% of sterols in the standard ST library were used as a training data set, and the remaining 43% served as an external validation data set. The external validation results demonstrated that the CCS prediction had a good linear fit with an $R^2$ value of 0.8690 and a median relative error (MRE) of 1.75% (Fig. 3b and Supplementary Data 4). Similarly, the external validation results also demonstrated that the RT prediction had a good linear fit with an $R^2$ value of 0.9813 and a median error (ME) of 22 s (Fig. 3c and Supplementary Data 4). Dimension 4 was the MS/MS spectral library predicted using the fragmentation rule. Previous studies have demonstrated that picolinyl derivatized sterols favor the generation of characteristic fragment ions under collision-induced dissociation (CID)-based fragmentation[23], such as the sterol skeleton ion $[M + H–PA]^+$ for monohydroxysterols, $[M + Na–PA]^+$ for dihydroxysterols, and the derivate ion $[PA + Na]^+$. We predicted the related characteristic fragments for all sterols in the extended ST library. The four-dimensional information for 14-demethyllanosterol is given as an example in Fig. 3d. Collectively, we curated the four-dimensional sterol library including

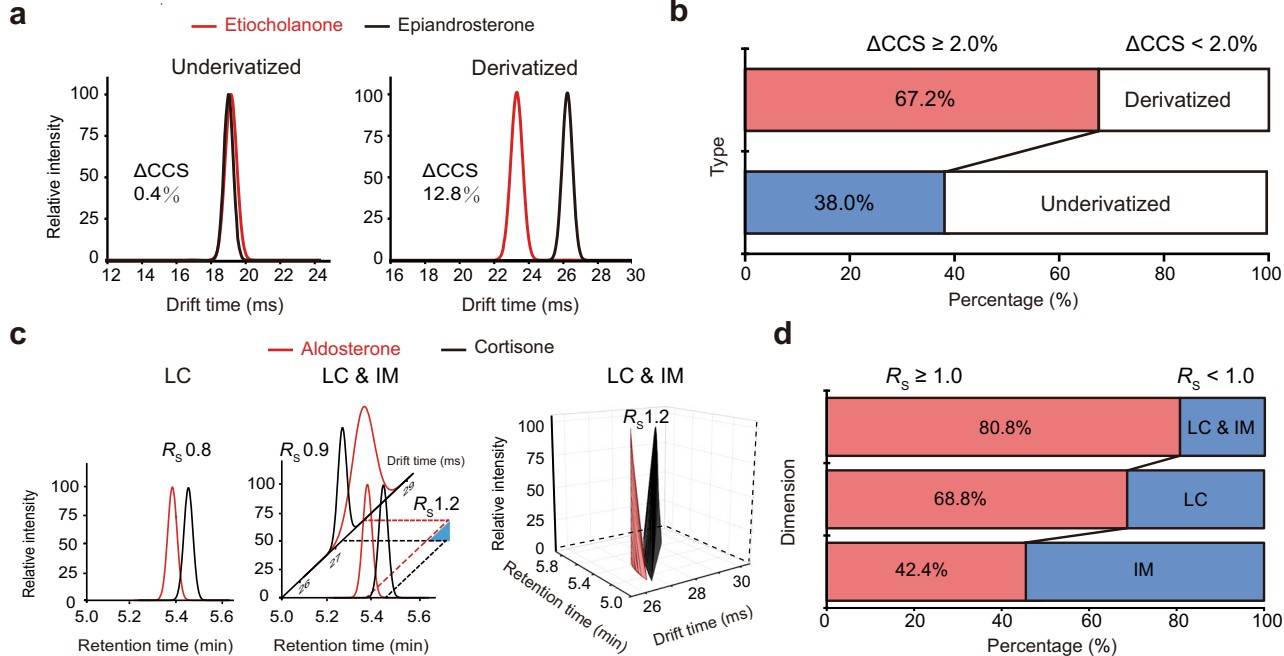

**Fig. 2 Improved separation of sterol isomers with derivatization and IM-MS. a** Overlay of the IM mobiligrams of underivatized and derivatized sterol isomers of etiocholanone and epiandrosterone. **b** The percentages of sterol isomers with significant CCS differences (≥2%) in the derivatized (*n* = 125) and underivatized (*n* = 163) groups. **c** The peak resolutions (*R*$_S$) in LC separation (left), IM separation and LC-IM based two-dimensional separation (middle and right) for the sterol isomers of aldosterone and cortisone after derivatization. **d** The percentages of sterol isomers (*n* = 125) with peak resolutions larger than 1.0 in LC separation, IM separation, and LC-IM based two-dimensional separation.

both a standard library and an extended library to support the IM-MS-based comprehensive analysis of sterol lipids.

**IM-MS-based four-dimensional analysis of sterol lipids**. We demonstrated the application of IM-MS technology for the four-dimensional analysis of sterol lipids in complex biological matrices. In total, 278 sterol lipids were identified using the four-dimensional analysis of sterol lipids in common biological samples (Fig. 4a, b and Supplementary Data 5 and 6). These sterol lipids were from 4 main classes and 19 subclasses. The detailed analysis revealed that ST01 sterol was the dominant class with high proportions classified as cholesterol and derivatives (31%), ergosterols and derivatives (18%), and stigmasterols and derivatives (15%). Among them, 20–21 sterols were identified using the standard ST library and further confirmed with chemical standards (Fig. 4b and Supplementary Fig. 3). The identification of campesterol using the standard ST library is given as an example in Fig. 4c. In brain tissue, six features were putatively annotated as campesterol with MS1 matching. The addition of RT matching, CCS matching, and MS/MS spectral matching was used to filter false positives and annotated the feature M528T849C269 as campesterol. This identification was validated using the chemical standard (Fig. 4d). Specifically, these sterol identifications were classified as level 1 according to the definitions of Metabolomics Standard Initiative (MSI)[42].

In addition, 65–176 sterols were identified by four-dimensional technology using the extended ST library (Fig. 4b). The sterol identifications from the four-dimensional matching with the extended ST library were classified as MSI level 2. The identification example of 14-demethyllanosterol using the extended ST library is given as an example in Fig. 4e. In brief, four features were putatively annotated as 14-demethyllanosterol with MS1 matching in brain tissue. The addition of RT matching, CCS matching, and MS/MS spectral matching with the extended ST library finally annotated the feature M540T795C268 as 14-

demethyllanosterol (Fig. 4e). To validate it, we further purchased the authentic standard of 14-demethyllanosterol and confirmed the identification result (Fig. 4f).

Previous reports have demonstrated that, in ion mobility separation, specific lipid species can be separated according to their different trend lines[34,36,39,43]. Therefore, trend line technology was further used to validate the sterol identified by the four-dimensional analysis of sterol lipids. First, we fitted a trend line for sterol lipids using a power function with the standard ST library ($y = 22.2x^{0.38}$, $R = 0.7149$; *n* = 97; Fig. 4g). Then, the fitted trend line was applied to all identified sterol lipids in plasma, liver, and brain tissue samples. The results demonstrated that all sterols fell within the sterol chemical space with a 99% predictive interval (Fig. 4h). Collectively, these results demonstrated that the IM-MS technology empowered by the four-dimensional library can support sterol lipid analysis with high accuracy and extended coverage.

**Spatial diversity of sterol lipids in the mouse brain**. The profiling of sterol lipids in the whole mouse brain is well reported[11,18], however, the distribution of distinct sterol lipids in different brain regions and the related functions remain largely unexplored. Therefore, we dissected ten functional regions of the mouse brains (olfactory bulb, anterior cortex, posterior cortex, hippocampus, cerebral nuclei, interbrain, midbrain, pons, medulla, and cerebellum) and quantitatively measured sterol lipids using the four-dimensional technology (Fig. 5a). In total, 197 sterol lipids from 14 subclasses were identified in the brain functional regions (Fig. 5b and Supplementary Data 6). Different numbers of sterol lipids were characterized in distinct brain regions, with the olfactory bulb being the most sterol-rich (Fig. 5b). Further cross-regional analysis revealed the diversity of sterol distributions among different brain regions. We found that sterol lipids had diverse landscapes in terms of both numbers and concentrations across the brain spatially. Of note, 51 sterol lipids in class ST01 were present in all brain regions measured, which

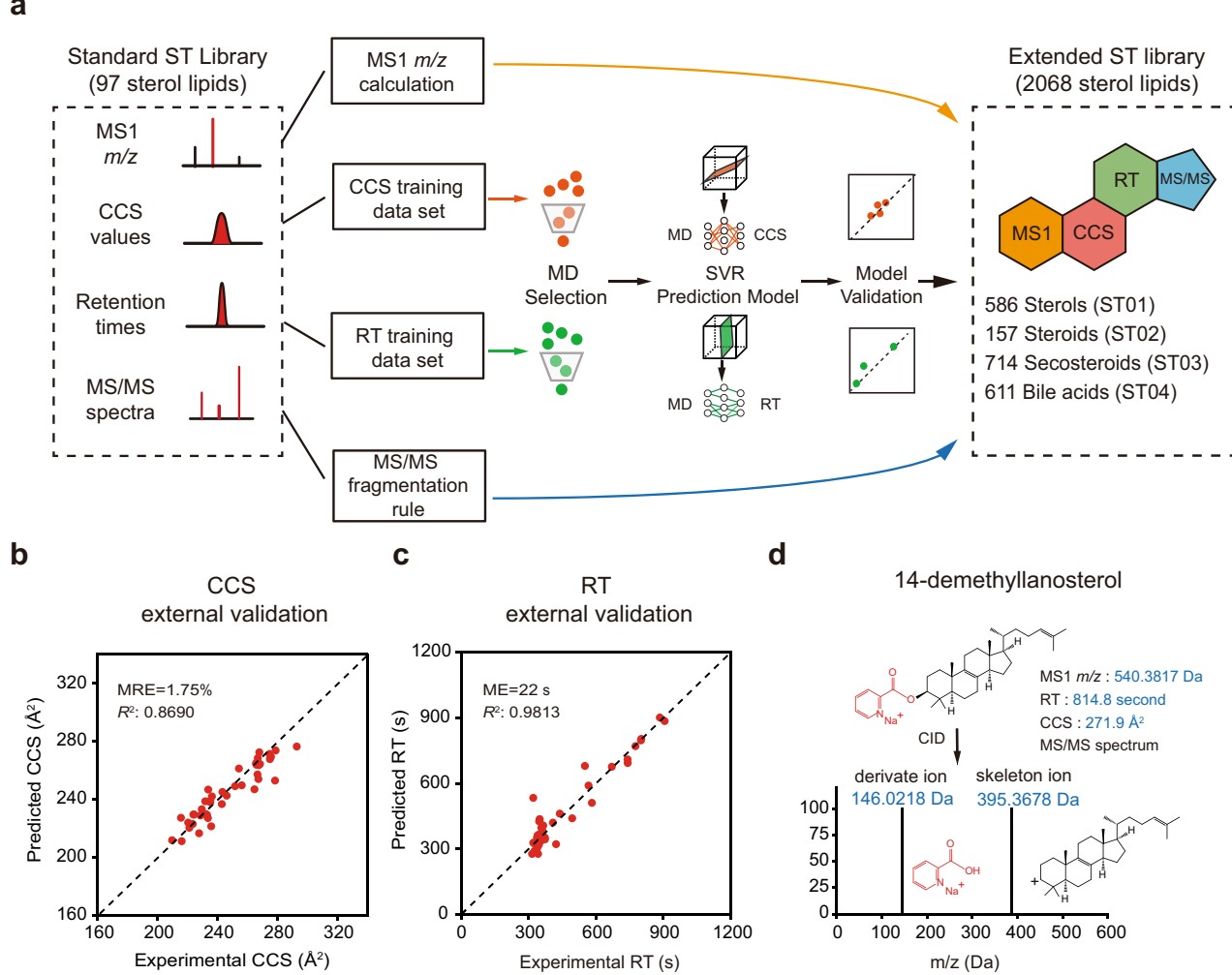

**Fig. 3 Machine learning-enabled four-dimensional library for the analysis of sterol lipids. a** Schematic illustration of the workflow to curate the standard and extended ST libraries with four-dimensional information. **b** The external validation of the machine-learning algorithm predicted CCS values of sterols ($n = 42$). **c** The external validation of the machine-learning algorithm predicted retention times (RTs) of sterols ($n = 42$). **d** An example of four-dimensional information of 14-demethyllanosterol in the extended ST library.

indicates the essential roles of these sterols in normal physiological activities in the whole brain (Fig. 5c and Supplementary Data 7). As a comparison, secosterol-B was uniquely detected in the anterior cortex, while 24-isopropyl-cholesterol and iso-fucostanol were present in only the midbrain and interbrain but not in the other functional brain regions (Supplementary Fig. 4).

Then, we examined the concentration distributions of sterol lipids measured in each brain region (Supplementary Data 8–9). High concentrations of cholesterol in the brain have been well documented. Indeed, we found that cholesterol was the most abundant sterol lipid in all brain regions with a minimum concentration of 3.83 ($\pm 0.78$, s.d.) $\times 10^3$ ng/mg in the anterior cortex and a maximum value of 2.72 ($\pm 2.35$, s.d.) $\times 10^4$ ng/mg in the pons region. The concentration ranges of all sterol lipids in different brain regions were vast, reaching up to 8 orders of magnitude. The least abundant sterol lipid detected in this study was 1,17-dihydroxy-androstan-3-one in the olfactory bulb (1.84 ($\pm 0.6$, s.d.) $\times 10^{-4}$ ng/mg), while the most abundant sterol lipid detected was cholesterol in the pons (2.72 ($\pm 2.35$, s.d.) $\times 10^4$ ng/ mg). Interestingly, the concentrations of 22-hydroxy-23,24,25,26,27-pentanorvitamin D3 spanned the widest range in all brain regions (from 9.34 ($\pm 3.08$, s.d.) $\times 10^{-4}$ ng/mg in the hippocampus to 0.423 ($\pm 0.052$, s.d.) ng/mg in the cerebellum by

453-fold difference). Further stratification by sterol subclasses showed that the top three most abundant subclasses across the brain were cholesterol derivatives, ergosterol derivatives, and stigmasterol derivatives (Fig. 5d). Of note, the concentrations of the subclass cholesterol derivatives spanned the widest range in all brain regions (Supplementary Fig. 5). Spirostanol derivatives were found to have the lowest concentration in seven out of ten brain regions (Fig. 5d and Supplementary Fig. 6).

To comprehensively understand the differences in sterol lipids between brain regions, univariate analysis was carried out using one-way ANOVA, and 149 sterol lipids were found to be statistically altered ($p$-value <0.05). With these 149 altered sterol lipids, hierarchical clustering analysis (HCA) was performed to investigate the variation and similarities of sterol lipids among individual brain regions. As a result, five cluster groups were generated in an HCA-based heat map plot (Fig. 5e and Supplementary Data 10). Interestingly, three cluster groups contained the dominant classes of sterol lipids. For example, cluster 1 mainly consisted of oxysterols (53%), and these oxysterols were enriched in the cerebral nuclei and olfactory bulb but were least abundant in the cerebellum (Fig. 5f). This finding is consistent with a previous report on oxysterols[15]. Remarkably, cluster 2 was dominated by phytosterols (62%),

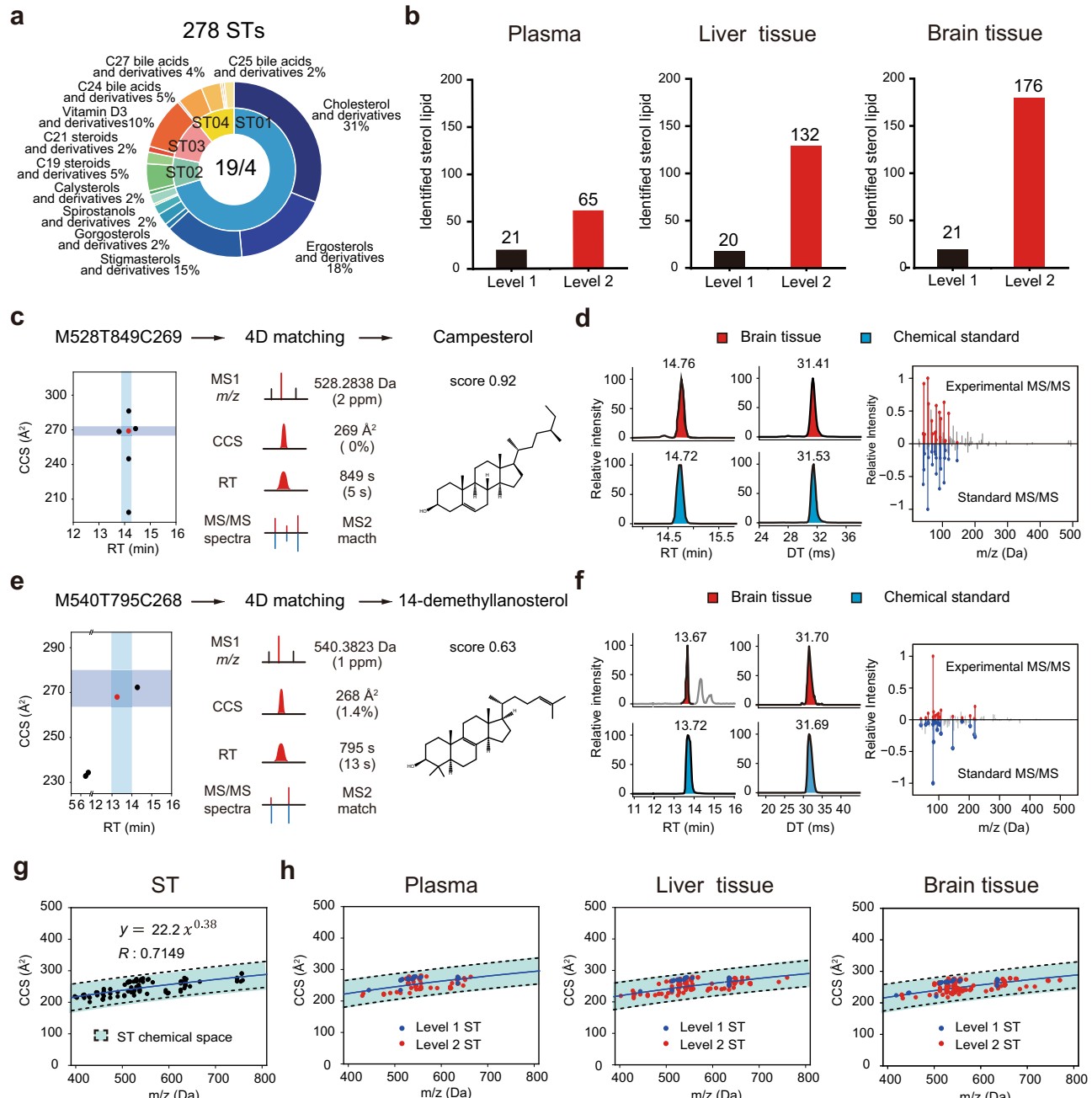

**Fig. 4 IM-MS-based four-dimensional sterol lipid analysis of biological samples. a** Subclass statistical analyses of the identified sterol lipids in human plasma ($n = 3$ technically replicated samples), mouse liver tissue ($n = 6$ biologically independent samples), and mouse brain tissue ($n = 6$ biologically independent samples for each group). Each sample was analyzed once by LC-IM-MS. **b** Numbers of identified sterol lipids with MSI level 1 and MSI level 2 in three sample types. **c** The identification of campesterol in brain tissue samples using the standard ST library through four-dimensional matching. **d** Validation of campesterol identification with the chemical standard. **e** The identification of 14-demethyllanosterol in brain tissue sample using the extended ST library through four-dimensional matching. **f** Validation of the 14-demethyllanosterol identification with the chemical standard. **g** Fitted trend line and chemical space for sterol lipids as determined using a power function with the standard ST library ($n = 97$). **h** The distributions of the identified sterol lipids from three sample types in the chemical space of sterol lipids. Blue solid line, the trend line of sterol lipids. Black dash lines, the 99% predictive interval.

which were expressed at higher levels in the medulla and the pons than in the other brain regions (Fig. 5g). Cluster 3 was a mixture of cholesterol derivatives and other sterols. The cholesterol derivatives exhibited the highest levels in the pons but had lower levels in the anterior cortex and cerebellum (Fig. 5h). Specific examples of oxysterols, phytosterols, and cholesterol derivatives from clusters 1–3 are given in Supplementary Fig. 7. Altogether, these results demonstrated that different brain regions have distinct spatial profiles of sterol

lipids. The region-specific distribution of sterol lipids suggests that the diverse functions of brain regions may require different sterol metabolisms.

**Age-associated spatial diversity of sterol lipids in the mouse brain.** Abnormalities in sterol metabolism have been implicated in different neurological diseases that are closely related to age. We next used four-dimensional technology to comprehensively

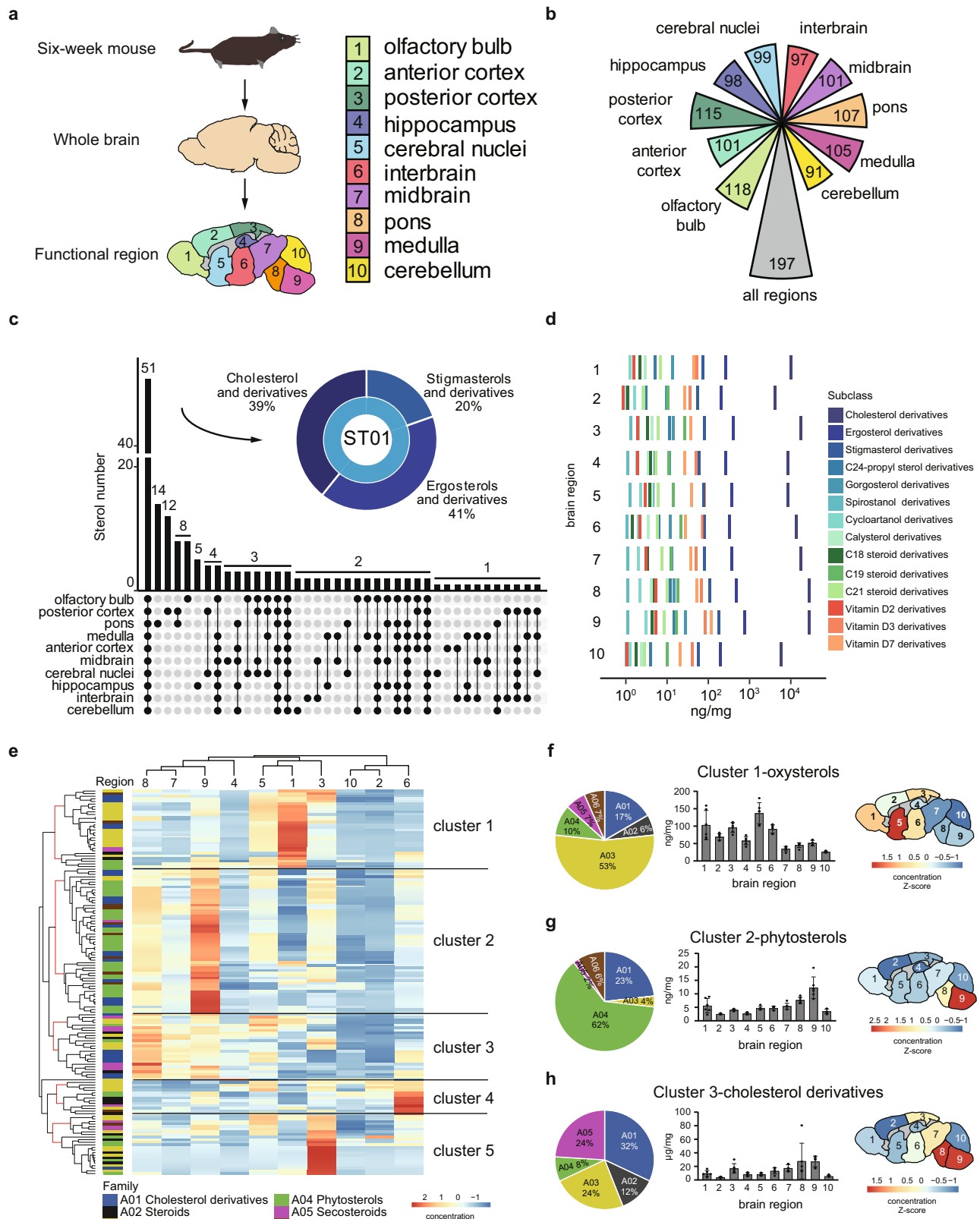

**Fig. 5 Spatial characterizations of sterol lipids in the mouse brain. a** Dissection of functional regions of mouse brains (6-week; *n* = 6 biologically independent samples for each group). Each sample was analyzed once by LC-IM-MS. **b** Numbers of sterol lipids identified in the brain regions. **c** The distributions of shared and unique sterol lipids in the brain regions. Each column represents the number of sterol lipids identified in the specific brain regions, which were shown as black solid dots in the lower part of the plot. **d** Absolute concentrations of sterol lipids in 14 sterol subclasses measured in the brain regions. The sterol concentrations were normalized to tissue wet weight (mg). **e** Hierarchical clustering analysis of sterol lipids in mouse brain regions (*n* = 149). **f–h** Percentages of sterol subclasses in clusters 1–3 and concentration distributions of representative sterol subclasses in mouse brain regions. Data are presented as mean values +/− SD (*n* = 6 biologically independent samples for each group).

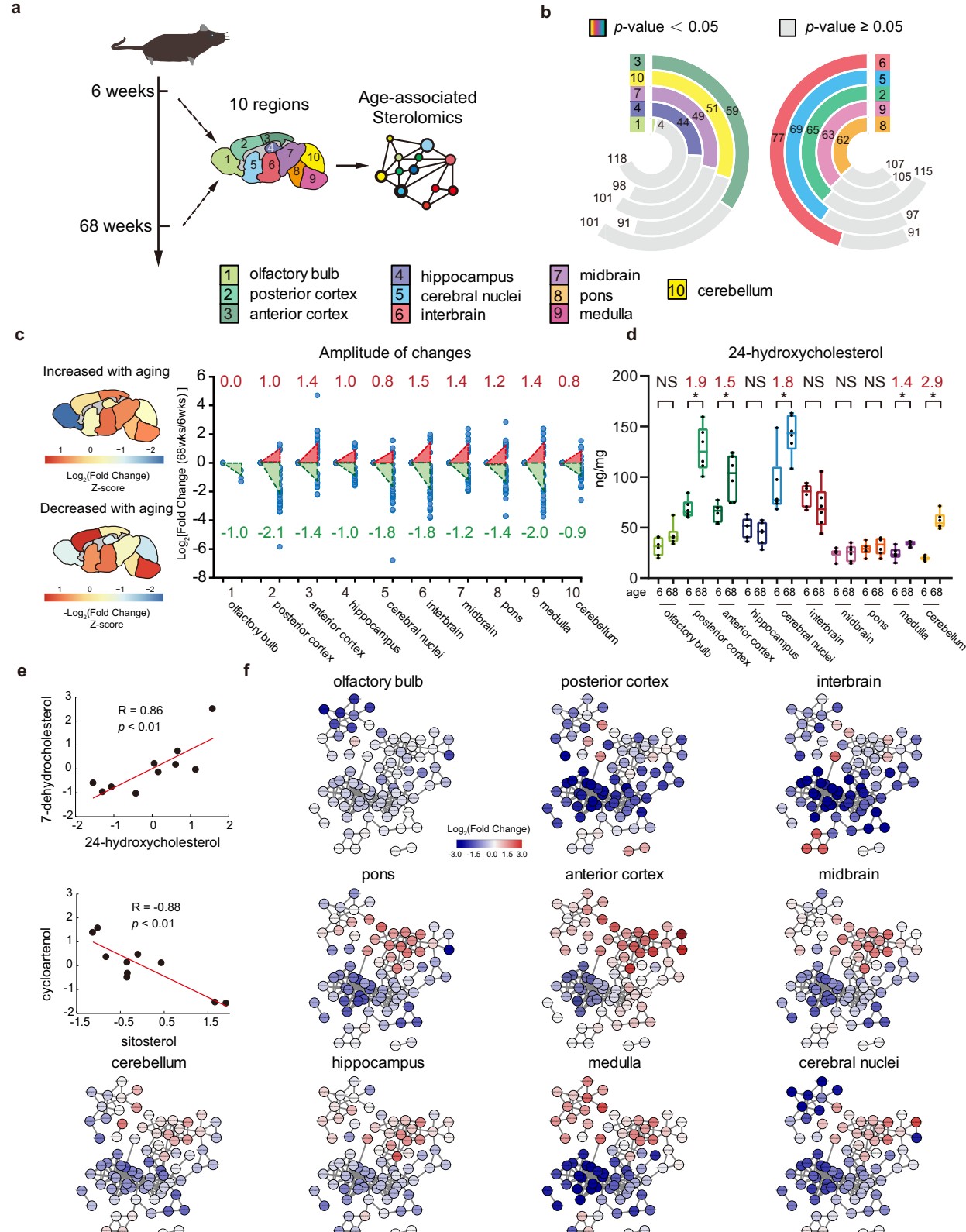

and spatially delineate the age-associated sterol lipid differences in the mouse brain. We quantitatively analyzed ten brain regions from mice at the ages of 6 and 68 weeks. We found that 102 sterol lipids identified in six mouse samples had significant age-associated differences in at least one brain region (*p*-value < 0.05, two-sided Student's *t*-test, Figs. 6a, b and Supplementary Data 11). Further statistical analysis of individual brain regions

revealed distinct changes in sterol lipids in response to aging. First, different numbers of significantly changed sterol lipids in different brain regions were observed. For example, 77 sterol lipids were significantly changed in the interbrain during aging while only four sterols were altered in the olfactory bulb (Fig. 6b). Furthermore, the amplitudes of age-associated changes in sterol lipids showed spatial diversity in the brain, and the change

**Fig. 6 Age-associated spatial diversity of sterol lipids in the mouse brain. a** Dissection of functional regions of the mouse brain (6 and 68 weeks old; $n = 6$ biologically independent samples for each group). Each sample was analyzed once by LC-IM-MS. **b** Numbers of significantly changed sterol lipids in brain regions between mice of two ages (two-sided Student's $t$-test; $p$-value adjusted using FDR). **c** Dysregulated amplitudes of sterol lipids in the brain regions of young and old mice. The mean $\log_2$ (fold-changes) values of dysregulated sterol lipids are marked in the plot. Red areas and green areas in the right panel represent mean increased and decreased amplitudes in the brain regions of young and old mice, respectively. **d** Concentrations and fold-changes of 24-hydroxycholesterol in the brain regions of young and old mice ($n = 6$ biologically independent samples for each group, two-sided Student's $t$-test; $p$-value adjusted using FDR; $p$-value <0.05: *; $p$-value ≥0.05: NS). Box plots display median value (centerline), upper and lower quartiles (box limits), 1.5× interquartile range (bar), and points out of interquartile range are outliers. **e** Scatter plots showing two example pairs of sterol lipids with positive and negative correlations over ten brain regions. The $p$-value of the upper example pair is 0.00537 two-sided Student's $t$-test, 95% confidence interval). The $p$-value of the lower example pair is 0.000721 (two-sided Student's $t$-test, 95% confidence interval). The red lines indicate the linear fitting. The x-axes and y-axes are standardized as $Z$-scores of the mean $\log_2$ (fold changes) values of sterols. **f** Network visualization of the sterol lipid correlations in ten brain regions between young and old mouse brains. The edges are correlations in the ten brain regions with $R \geq 0.85$ (Pearson correlation). The nodes are sterol lipids. The nodes of the network are color-coded based on the $\log_2$ (fold changes) values of sterols in each brain region.

patterns induced by aging in every brain region were dual-directional (Fig. 6c and Supplementary Fig. 8). The interbrain had the largest variation in the increased sterol lipids during aging, but no increased levels of sterol lipids were observed in the olfactory bulb. All ten brain regions had decreased levels of sterol lipids in older mice, with the change amplitude ranging from 0.9 in the cerebellum to 2.1 in the posterior cortex ($\log_2$ of fold change; Fig. 6c). Moreover, 24-hydroxycholesterol (24-HC), which is a dominant form of cholesterol that effluxes by crossing the blood-brain barrier, was found to be enriched in the cerebral nuclei and interbrain but had the lowest level in the cerebellum at baseline in six-week-old mice (Fig. 6d). Importantly, these findings are in line with the distribution of the synthetase of 24-HC enzyme CYP46A1 across brain regions[15]. In the brains of old mice, the levels of 24-HC were elevated significantly in the posterior cortex, anterior cortex, cerebral nuclei, medulla, and cerebellum.

Given the broad age-associated changes in sterol lipid profiles, we next sought to investigate the coregulation of sterol levels over ten different brain regions. Initially, Pearson correlation analyses were performed using the fold changes between 68-week-old and 6-week-old mice across all ten brain regions, and we found 220 pairs of positively correlated sterol lipids and 3 pairs of negatively correlated sterol lipids (Supplementary Data 12). For example, 24-hydroxycholesterol was positively correlated with 7-dehydrocholesterol between young and old mice in all ten brain regions (Fig. 6e). Exogenous plant sterol lipids, sitosterol, and cycloartenol displayed negative correlations between young and old mice across the ten brain regions (Fig. 6e). With a larger number of positive correlations between age-associated sterol lipids being observed, we next constructed a network for each brain region where nodes were sterol lipids and edges represented correlations of 0.85 or higher (Fig. 6f). Interestingly, the networks for the anterior cortex, hippocampus, cerebral nuclei, midbrain, pons, and medulla displayed bimodal separation, with sterol lipid increased in the old mouse brain were in the top right corner and decreased sterol lipids were at the bottom left side. The sterol lipids with increased levels in the brains of old mice in the network were mainly ergosterol derivatives (33.3%) and cholesterol derivatives (33.3%), while those with decreased levels were enriched with cholesterol derivatives (43.6%). This bimodality suggested distinct sterol lipid metabolism associated with age in related brain regions. However, this characteristic finding was not observed in the olfactory bulb, posterior cortex, or interbrain, where the majority of coregulated sterol lipid exhibited decreased levels in the brains of old mice. Strikingly, the levels of the coregulated sterol lipids were increased in the old mouse brain in the anterior cortex and were mostly decreased in the posterior cortex despite the similar physiological structures between the two brain regions (Fig. 6f). In summary, our results delineated the

spatial and temporal diversity of sterol lipids in the mouse brain and revealed the implication of interplay between age-associated sterol lipids in each brain region. Furthermore, we developed an interactive web-based application for researchers to explore the levels of different sterol lipids across distinct brain regions (http://mousebrainatlas.zhulab.cn/).

## Discussion

The biological importance of sterol lipids particularly in the brain is widely recognized. However, the analysis of sterol lipids in the brain has been largely restricted by coverage. In most studies, only a few sterol lipids were detected in the brain because of the unavailability of sterol standards and related databases[11,18,44]. In addition, the differentiation of sterol isomers using mass spectrometry is a well-recognized challenge. In this work, we first demonstrated that picolinyl derivatization together with ion mobility separation can largely improve the separation of sterol isomers. This improvement stems from the enlarged differences in the rotationally averaged surface areas of derivatized isomers under the scenario of ion mobility separation. Although some studies have reported the use of IM-MS for sterol lipid analysis[33,36,45,46], in-depth characterization of sterol lipids on a large scale with IM-MS has not been achieved due to the lack of a comprehensive four-dimensional database for sterol lipids. In this study, we first developed a high-coverage IM-MS-based technology empowered by a machine learning-enabled four-dimensional library. The four-dimensional library included information on $m/z$, CCS, RT, and MS/MS spectra for 2165 sterol lipids. In complex biological samples, we were able to identify 278 sterol lipids from 14 sterol subclasses with high accuracy and extended coverage.

Difficulties in sterol lipid identification have made the comprehensive characterization of sterol lipids in biological samples challenging. Our IM-MS-based technology also has limitations in sterol identifications. First, due to the limitation of picolinyl derivatization, some important sterols such as cholesterol ester (CE) and steroids without hydroxyl groups were missing from the current analysis. Second, the limited number of available sterol standards impeded the accuracy of predicted CCS values in the extended ST library. The low accuracy of predicted CCS values is prone to generate false sterol identifications. To minimize the effect, we employed the experimental CCS values of the standard ST library for the identification of level 1 sterol lipids. However, for level 2 identifications, the false identifications are unavoidable since the predicted CCS values were used for matching. To examine the identification accuracy using the extended ST library, we estimated the false discovery rate (FDR) of sterol lipids in the olfactory bulb sample which is the sterol-richest brain region. With the addition of CCS matching for sterol lipid identification, the FDR decreased dramatically (Supplementary Fig. 9). The estimated FDR value in the data set was 1.4% when the CCS match tolerance was set as 3%. Thus,

the results showed that our four-dimensional method enabled the characterization of sterol lipids in biological samples with high annotation accuracy. In the future, with the increase of available sterol standards, we could gradually expand the size of the training data set. We could also develop alternative machine-learning algorithms such as neural network algorithm to further improve the predicted accuracy of CCS values.

The comprehensive profiles of sterol lipids in ten brain regions deciphered in this study clearly revealed that sterol lipids were unevenly distributed across the brain. This was evidenced by two distinct layers of our findings. First, different sterol lipids were identified in different brain regions. For example, secosterol-B was present solely in the anterior cortex in the mouse brain. Notably, secosterol-B levels were shown to be increased in atherosclerotic plaques[47] and in the brain tissues of patients with AD[48]. Although the relationship between secosterol-B and the pathogenesis of AD has not been well studied[49,50], it is plausible that the accumulation of secosterol-B being unique to the anterior cortex could promote disease development. Second, for the sterol lipids commonly detected in distinct brain regions, their concentrations were differentially quantified. Among these sterol lipids, the most salient discovery was the characterization of phytosterol as a dominant class that was highly abundant in the medulla and pons. Incidentally, the medulla and pons are two unique brain regions that have been indicated to be related to the pathophysiology of PD[51–53]. As phytosterols can be gained only from the diet, their levels in the brain are reflective of the intestinal absorptive capacity[8,54]. Further investigation of the relevance of brain region-specific phytosterol enrichment and intestinal absorption and transduction will be of utmost importance to elucidate the unsolved pathogenesis of PD[51–53,55].

A decline in brain functions is a characteristic feature of aging, and this process is usually accompanied by dysregulation of sterol lipid metabolism[10,18,19,44]. Our data revealed how sterol lipids are affected by age in a brain region-specific manner, however, relatively few mouse samples were investigated. The interbrain was found to be the most deeply affected brain region, exhibiting the largest number of changed sterol lipids and the most extensive amplitude variation. In contrast, the olfactory bulb region of the mouse brain was less vulnerable to age. This difference might be related to the distinct functions of the two brain regions and revealed that the diverse functions of different brain regions require different sterol metabolisms. Another notable finding of this study is the coregulation of sterol lipids across ten brain regions as revealed by the network analysis. As an enzyme in sterol lipid metabolism can be responsible for multiple sterol lipids[17,56], these sterol lipids with tight coregulation can feasibly act in a coordinated manner across the brain to respond to aging. Despite being coregulated, the age-associated changes in sterol lipid levels were spatially different. This was supported by the elevated levels of coregulated sterol lipids in older mice in the anterior cortex, hippocampus, cerebral nuclei, midbrain, pons, and medulla regions but decreased levels in the olfactory bulb, posterior cortex, and interbrain regions of older mice. It is conceivable that the coregulatory networks are brain region-dependent since different functions within distinct regions are underpinned by sterol lipid metabolism. It is worth noting that although we have carefully followed the protocol of performing the dissection experiments at a low temperature and rapidly freezing the samples during collection to minimize postmortem changes in sterol lipids that may occur in brain tissues[57], the possible oxidation of sterol lipids is unavoidable. This fact may have contributed to the sterol profiles reported in this work being partially dependent on the protocol used to harvest the brains and the impact on sterol profiles.

## Methods

**Chemicals**. Chemical standards of sterol lipids were purchased from J&K Scientific (Shanghai, China), Avanti Polar Lipids (Alabaster, USA), Steraloids (Newport, RI, USA), and Sigma-Aldrich (St. Louis, MO, USA). Acetic acid was purchased from Fisher Scientific (Morris Plains, NJ, USA). Picolinic acid, pyridine, and 2-methyl-6-nitrobenzoic anhydride (MNBA) were purchased from J&K Scientific (Shanghai, China). 4-dimethylaminopyridine (DMAP) was purchased from Sigma-Aldrich (St. Louis, MO, USA). Triethylamine (TEA) was purchased from China National Pharmaceutical Group Corporation (Shanghai, China). Potassium hydroxide was purchased from BBI Life Science Corporation (Shanghai, China). Butylated hydroxytoluene (BHT) was purchased from Sigma-Aldrich (St. Louis, MO, USA). LC-MS grade water ($H_2O$), acetonitrile (ACN), methanol (MeOH), HPLC grade dichloromethane (DCM), and HPLC grade hexane were purchased from Honeywell (Muskegon, MI, USA).

**Sample preparation**. Sample preparation includes mouse brain dissection, sterol extraction, hydrolysis, derivatization, and sample cleaning. Mice (C57BLJ6; female; 6-week and 68-week, $n = 6$ biologically independent samples for each group) were first anesthetized with isofluorance vapor and sacrificed by decapitation. The whole brain was manually divided into different subregions by referring to the mouse brain atlas (Paxinos and Franklin, The Mouse Brain, 4th edition). The brain tissue was kept in ice-cold dissection buffer (212.7 mM sucrose, 5 mM KCl, 1.25 mM $NaH_2PO_4$, 10 mM $MgCl_2$, 0.5 mM $CaCl_2$, 26 mM $NaHCO_3$, and 10 mM dextrose, and bubbled with 95% $O_2$/5% $CO_2$ (pH 7.4)) during the whole process to prevent neural excitotoxicity and maintain viability. The tissue was directly frozen in liquid nitrogen and kept at −80 °C before the further experiment. It took ~15 min to fully dissect one mouse brain immediately after euthanasia. All animal experiments were approved by the Institutional Animal Care and Use Committee (IACUC) of Interdisciplinary Research Center on Biology and Chemistry, Shanghai Institute of Organic Chemistry, Chinese Academy of Sciences, and comply with all relevant ethical regulations. The 6-week and 68-week old female mice were purchased from Xiamen University Laboratory Animal Center (Xiamen, China). All mice were maintained on a 12-h light/dark cycle at room temperature (24 ± 2 °C) with constant humidity (40 ± 15%).

Extraction of sterol lipids followed the method published by our group[23]. In brief, the frozen brain tissue was weighed and homogenized in $H_2O$ (every mg of tissue with 20 μL of $H_2O$) using a homogenizer (Precellys 24, Bertin Technologies, France). The whole homogenization process was operated under liquid nitrogen cooling. Ten microliters of butylated hydroxytoluene (BHT) solvent (6.5 μg dissolved in 10 μL of MeOH) were added into the sample before homogenization to avoid possible oxidation during sample preparation. Then, 100 μL of the homogenized solution was taken for each sample and mixed individually with 30 μL of deuterated internal standards solution (IS; 10 ng of d6-desmosterol, 10 ng of d7-cholesterol, and 10 ng of d6-27-hydroxycholesterol), and further diluted to 200 μL by adding $H_2O$. Eight hundred microliter of the extraction solvent DCM/MeOH (2:1; v/v) containing 6.5 μg BHT was added for extraction. The solution was vortexed for 30 s, followed by 10 min of sonication, and 15 min of centrifugation at 900×$g$ at 4 °C. The bottom organic layer was collected (400 μL). An additional 400 μL of DCM was added to the rest layer for re-extraction. The extraction was repeated three times in total. The pooled organic layer was evaporated using a vacuum concentrator (LABCONCO) at 4 °C.

For hydrolysis, the dried extract was mixed with 500 μL of a methanol solution containing 1.0 M potassium hydroxide and 6.5 mg of BHT. The solution was vortexed for 30 s, sonicated for 10 min, and incubated at 37 °C for 1 h. Then, 1 mL of hexane was added for extraction. The sample was vortexed for 30 s, sonicated for 10 min in a 4 °C water bath, and centrifuged for 15 min at 900×$g$ at 4 °C. The resulting supernatant was collected (600 μL). For the second extraction, 1.0 mL of hexane was added, and 1.0 mL of supernatant was taken and combined with supernatant in the first extraction, and finally evaporated to dryness using a vacuum concentrator at 4 °C.

Sterol derivatization was performed according to the previous method with minor modification[23]. The chemicals for derivatization including picolinic acid (53.3 mg), 2-methyl-6-nitrobenzoic anhydride (66.7 mg), and 4-dimethylaminopyridine (3 mg) were sequentially added to pyridine (1 mL). The freshly prepared derivatization reagent (200 μL) and 5.36 μL triethylamine were added to the dried extract. The reaction mixture was vortexed for 30 s, and sonicated for 10 min, and incubated at 80°C for 60 min. Then 1 mL of hexane was added to samples for extraction. The samples were vortexed for 30 s, sonicated for 10 min (4 °C water bath), and centrifuged for 15 min at 900×$g$ at 4 °C. Finally, the supernatant was collected and evaporated to dryness using a vacuum concentrator at 4 °C. The dry extract after derivatization was mixed with 400 μL of dichloromethane and 600 μL of $H_2O$ for sample cleaning. Then the solution was vortexed for 30 s, sonicated for 10 min in a 4 °C water bath, and centrifuged for 15 min at 900×$g$ at 4 °C. Next, 400 μL of the upper aqueous layer was removed. Then, another 400 μL of $H_2O$ was added to the rest sample solution for cleaning. The cleaning step was repeated twice. The rest organic layer was collected and evaporated to dryness using a vacuum concentrator at 4 °C. The dried extract was kept at −80 °C and reconstituted using 100 μL of ACN prior to LC-IM-MS analysis. For the analysis of cholesterol, the samples were first diluted by 50 times using ACN before analysis.

For the preparation of mouse liver tissue samples, the mice were anesthetized by isoflurane vapor before sacrifice (C57BLJ6; $n = 6$; 24-week), and livers were quickly taken, and dissected on ice. Dissected livers were immediately frozen in liquid nitrogen and stored at −80 °C for subsequent extraction of sterol lipids. The mice in this experiment were purchased from Shanghai Model Organisms Center (Shanghai, China). The frozen liver tissue was weighed and homogenized. Then, 50 μL of the homogenized solution was taken for sample preparation. The rest procedures were the same as the preparation of brain tissues. For the preparation of human plasma, 10 μL of plasma were used for sterol extraction. The procedures were the same as the preparation of brain tissues. Human plasma was purchased from Equitech-Bio (Catalog No. HPH-0500, TX, USA).

**LC-IM-MS/MS analysis**. The LC-IM-MS/MS analyses were performed using an Agilent DTIM-QTOFMS 6560 coupled with an Agilent UHPLC 1290 (Agilent Technologies, USA). Chromatographic separations were performed on a Phenomenex Kinetex C18 column (particle size, 1.7 μm; 100 mm (length) × 2.1 mm (i.d.)) with column temperature maintained at 50 °C. The mobile phases (A, water with 0.1% acetic acid; B, methanol with 0.1% acetic acid) were used for gradient separation. The elution gradient and flow rates were set as follows: 0–3 min: 0% B to 25% B, 3–5 min: 25% B to 85% B, with the flow rate of 0.4 mL/min; 5–5.1 min, isocratic step at 85% B, with the flow rate increasing to 0.6 mL/min; 5.1–14 min, 85% B to 93% B, with the flow rate of 0.6 mL/min; maintained for 3 min, then ramp to 100% B within 0.2 min, and kept for 1.8 min; returned to initial conditions within 0.1 min; maintained 0% B during 19.1–19.2 min with the flow rate decreased to 0.4 mL/min; equilibrating the column with the initial condition for 0.8 min. The total gradient time was 20 min. The sample was maintained at 4 °C during the whole analysis. The MS parameters were set as the follows: ESI positive mode; mass range, 124 to 1300 Da; sheath gas temperature, 350 °C; sheath gas flow, 12 L/min; dry gas temperature, 365 °C; drying gas flow, 8 L/min; nebulizer pressure, 20 psi; capillary voltage, 3500 V. The CCS values were measured using the single-field method with the nitrogen as the drift gas. The maximum drift time was set as 60 ms and the scan rate was set to 0.9 frames per second. The pressure of the drift tube was set at 3.95 Torr, and the temperature of the drift tube was set as 300 K. The entrance and exit voltages of the drift tube were set as 1700 V and 250 V, respectively. Trap filling and trap release times were set as 20,000 μs, and 150 μs, respectively. The MS/MS spectra were acquired in the "Alternating frames" mode. The collision energy was set as 0 V in frame 1. The collision energy was set as 20 V in frame 2. The "targeted MS/MS" mode was used for the targeted MS/MS acquisition. All data acquisitions were carried out using MassHunter Workstation Data Acquisition Software (Version B.08.00, Agilent Technologies, USA). A retention time quality control (RTQC) was run with each batch to monitor RT shift and used for RT calibration (Supplementary Fig. 10 and Supplementary Table 1).

**Curation of the standard ST library**. We used the aforementioned LC-IM-MS/MS method to acquire the RTs, CCS values, and MS/MS spectra of 97 derivatized sterol standards. RTs and CCS values of sterol standards were acquired by the "Alternating frames" mode. MS/MS spectra of sterol standards were acquired by targeted MS/MS acquisition (CE = 20 V). The spectrum with the highest intensity was retrieved and normalized as the standard spectrum in the standard ST library. The standard ST library included calculated MS1, RTs, CCS values, and MS/MS spectra of 97 sterol lipids (Supplementary Data 3).

**Curation of the extended ST library**. To develop a machine-learning-based prediction of RTs and CCS values, we divided the 97 sterol lipids from the standard ST library into a training data set (3/5; $n = 55$) and an external validation data set (2/5; $n = 42$). The simplified molecular-input line-entry specification (SMILES) of 97 sterol lipids were collected from LIPID MAPS Structure Database [https://www.lipidmaps.org/data/structure/download.php] (accessed on December 16th, 2016) and PubChem [https://pubchem.ncbi.nlm.nih.gov/] (accessed on December 26th, 2016) to describe the chemical structures. Then, 221 molecular descriptors (MD) were calculated from SMILES by R package "rcdk" (version 3.3.8). Next, MDs were selected using the least absolute shrinkage and selection operator (LASSO) algorithm and the training set. We used 10-fold cross-validation to optimize "lambda" value in LASSO, and the "lambda" with the lowest mean squared error was selected as the best model. Finally, 12 and 24 MDs were selected for CCS and RT predictions. The R package "glmnet" (version 4.1) was used here. Then, the CCS and RT predictions were developed using the support vector regression (SVR) algorithm, which utilized a kernel function to construct high-dimensional regression between selected MDs and CCS/RT values in the training data set. The detailed workflow followed our previous publication of AllCCS[34]. Briefly, two hyperparameters, cost of constraints violation (C) and gamma (γ) were optimized from 77 parameter combinations via 10-fold cross-validation with 100 repeats. The optimized parameters were optimized as follows: C, 23, and 29 for CCS and RT predictions, respectively; γ, 0.1/12 and 0.25/24 for CCS and RT prediction, respectively. The internal validation for CCS and RT predictions were performed via leave-one-out (LOO) validation in the training data set (Supplementary Fig. 2). Finally, the SVR-based CCS and RT predictions were further validated using the external validation data set (Fig. 3b; $n = 42$).

To curate the extended ST library, we first retrieved a list of 2832 sterol lipids from the LIPID MAPS Structure Database [https://www.lipidmaps.org/data/structure/download.php] (accessed on December 16th, 2016), and removed 667 sterol lipids with 0 and multiple hydroxyl groups ($n > 5$). Among the 2,165 sterol lipids, 97 of them belong to the standard ST library, while the rest of 2068 sterol lipids were included in the extended library. MS1 $m/z$ values of 2068 sterol lipids were calculated using the formula after derivatization. The MS/MS spectra were generated by using the fragmentation rule. RTs and CCS values of 2,068 sterol lipids were generated by SVR-based prediction models. The extended ST library was provided in Supplementary Data 3.

**Data processing**. Raw MS data files (.d) were first recalibrated using IM-MS Reprocessor (Version B.08.00, Agilent Technologies). The smoothing and saturation repair was then performed using PNNL PreProcessor (Version 2018.06.02). The CCS calibration was performed by IM-MS Browser software (Version B.08.00, Agilent Technologies). The calibration coefficients (Single-Field; TFix and Beta) of calibrants (Agilent ESI-L Low Concentration Tuning Mix) were calculated and applied to all data files. The pre-processed data files were further processed using Mass Profiler (Version B.08.01, Agilent Technologies) for feature finding, alignment, and MS/MS spectra extraction. Finally, the MS1 peak table and MS/MS spectra (.CEF files) were exported. The parameters of feature finding were set as follows: maximum ion volume as a measure of abundance; ion intensity ≥100 counts; select common organic molecules as the isotopic model; set the charge state as 1–1. For the alignment of multiple samples, RT tolerance was set as ±(0.0% + 0.3 min), and mass tolerance was set as ±(15 ppm + 0.2 mDa). For statistics and filtering settings: missing sample: 0.001 abundance; feature filter: Q score ≥50; sample occurrence: frequency ≥50% in at least one group.

**Four-dimensional identification of sterol lipids**. Identification of sterol lipids was performed by four-dimensional matching with the standard and extended ST libraries by an in-house R package Sterol4DAnalyzer. First, the RT calibration was performed by analyzing retention time quality control (RTQC) samples. The RT calibration followed the method published by our group. The RTs of sterol lipids in both libraries are updated after the RT calibration. For four-dimensional matching against the standard ST library, the MS1 match tolerance was set as 25 ppm. Then, the trapezoidal function was used for scoring RT and CCS matches, which followed the method published by our group in IM-MS-based lipidomics (LipidIMMS Analyzer[40]) and metabolomics (AllCCS[34]). For the RT match, the minimum and maximum tolerances were set as 0 s and 12 s, respectively. For the CCS match, the minimum and maximum tolerances were set as 1% and 1.5%, respectively. The MS/MS spectral match was scored by a reverse *dot product* algorithm with a cutoff score of 0.6. A linear equation was used to integrate RT match, CCS match, and MS/MS spectral match scores. The weights of RT match score, CCS match score, and MS/MS match score were 0.2, 0.4, and 0.4, respectively. For four-dimensional matching against the extended ST library, the MS1 match tolerance was set as 25 ppm. For the RT match, the minimum and maximum tolerances were set as 0 s and 30 s, respectively. For the CCS match, the minimum and maximum tolerances were set as 1% and 3%, respectively. For MS/MS spectral match, if both skeleton ion $[M + H–PA]^+$ for monohydroxysterols, or $[M + Na–PA]^+$ for dihydroxysterols, and derivate ion $[PA + Na]^+$ were detected, the match score was set as 1.0. If only one type of fragment ion was detected, the match score was set as 0.5. The cut-off score for MS/MS match was set as 0.5. Similarly, a linear equation was used to integrate RT match, CCS match, and MS/MS spectral match scores. The weights of RT match score, CCS match score, and MS/MS match score were 0.2, 0.4, and 0.4, respectively. For four-dimensional matching against the standard and extended ST libraries, sterol identifications with integrated scores larger than 0.6 were kept.

**Validation of identified sterol lipids with chemical standard**. We used the aforementioned LC-IM-MS/MS method to acquire the RTs, IM drift times of 21 derivatized sterol standards, and brain samples. The "targeted MS/MS" mode was used for the targeted MS/MS acquisition, the collision energy was set as 60 V or 90 V.

**Quantification of sterol lipids in mouse brain tissues**. To quantify sterol lipids in brain tissue samples, 50 μL of homogenized solution were spiked with deuterated internal standards (ISs) before extraction, including d6-desmosterol (10 ng), d6-27-hydroxycholesterol (10 ng), and d7-cholesterol (10 ng). The calibration samples were prepared by mixing different concentrations of 21 sterol standards with three internal standards to generate the calibration curves. For the analyses of each brain region, calibration samples were analyzed two times at the beginning and end of the acquisition batch. The acquired data was processed by Skyline (version 19.1.0.193) with the import of a transition list including precursor $m/z$, precursor charge, RTs, and CCS values of identified sterols. The peak areas of sterol lipids were manually integrated, and chromatographic peak areas of identified sterols and three internal standards in biological samples were exported. Taken mouse cerebral nuclei region samples as examples, peak area ratios between 21 standards of sterol lipids and internal standards (sterol/IS) were calculated to generate 21 calibration curves (Supplementary Fig. 11 and Supplementary Table 2). The summary for calibration curves in other brain regions was provided in Supplementary Data 9.

The concentrations of sterol lipids with level 1 identifications in brain samples were directly interpolated from the calibration curves. The concentrations of sterol lipids with level 2 identifications in brain samples were interpolated from the calibration curves of sterol lipid standards within the same sterol subclass (Supplementary Table 3).

**Comparison of CCS differences and peak resolution for sterol isomers**. Two sterol lipids in the standard ST library with the same exact mass were regarded as one isomer pair. There are 125 and 163 pairs of sterol isomers in the derivatized and underivatized groups (Supplementary Data 1 and 2). The CCS difference provides a quantitative measure of the structural difference of two sterol lipids and is calculated by Eq. 1

$$CCS_{diff} = \frac{|CCS_A - CCS_B|}{CCS_A} \times 100 \qquad (1)$$

where $A$ and $B$ are sterol lipids with the same exact mass, $CCS_A$ is CCS value of $A$, $CCS_B$ is CCS value of $B$.

The resolution ($R_s$) provides a quantitative measure of the separation degree of two analytes[38]. For one-dimension peak resolution, such as LC separation, the resolution is calculated by Eq. 2:

$$R_{S_{LC}} = \frac{2(RT_A - RT_B)}{W_{RT_A} + W_{RT_B}} \qquad (2)$$

$RT_A$ is the retention time of $A$, $RT_B$ is the retention time of $B$, $W_{RT_A}$ is retention time peak width of $A$, $W_{RT_B}$ is retention time peak width of $B$.

Similarly, for IM separation, the resolution is calculated by Eq. 3:

$$R_{S_{IM}} = \frac{2|CCS_B - CCS_A|}{W_{CCS_A} + W_{CCS_B}} \qquad (3)$$

$CCS_A$ is CCS value of $A$, $CCS_B$ is CCS value of $B$, $W_{CCS_A}$ is peak width of $A$, $W_{CCS_B}$ is peak width of $B$.

For two-dimensional peak resolution of two orthogonal separations[37], the resolution is calculated by Eq. 4:

$$R_S = \sqrt{R_{S_{LC}}{}^2 + R_{S_{IM}}{}^2} \qquad (4)$$

where $R_{S_{LC}}$ is the resolution in LC separation, while $R_{S_{IM}}$ is the resolution in IM separation.

**Construction of trend line of the standard ST library**. The trend line was fitted using the nonlinear least-square function "nls" in R (version 4.0.2). The data was retrieved from the standard ST library ($n = 97$). The trend line was described as a power function $y = a \times x^b$, where, the $x$ is the precursor $m/z$ value of sterol lipid, and the $y$ is the precursor CCS value of sterol lipid. The starting point was set as ($a = 1$, $b = 0.05$). Finally, the trend line of sterol lipids was fitted as ($y = 22.2x^{0.38}$; $R = 0.7149$; see Fig. 4g). Second, the chemical space was constructed by the trend line with a 99% predictive interval (PI). The 0.99 PI was calculated by the following equation:

$$\triangle y(0.99PI) = Z \cdot S_{y,x} \cdot \left(1 + \frac{1}{n} + \frac{(x - \bar{x})^2}{SS_x}\right)^{1/2} \qquad (5)$$

$Z = 2.576$, which is the standard deviation $Z$-score based on 99% interval percentage;

$n = 97$, which is the number of data points (i.e., sterol lipid numbers);

$S_{y,x} = 17.94513$, which is the standard error of the $x$ and y data inputs;

$SS_x = 98.71926$, which is the sum of the squared deviations from the mean of $x$ inputs.

**Hierarchical clustering analysis**. The concentrations of sterol lipids in each brain region were first standardized to $Z$-scores. Hierarchical clustering was performed using an R package "pheatmap" (version 1.0.12) with default parameters. The mean values of each group were used for clustering calculation. The clustering method was the weighted Pair Group Method using Centroids (WPGMC). The correlation values were used as the clustering distance in the heatmap plot.

**Construction of coregulation network**. One hundred and two dysregulated sterol lipids were selected with a statistical difference ($p$-value <0.05; two-sided Student's $t$-test) in at least one brain region between young and old mouse brains from 197 sterol lipids. The fold changes (68-week vs. 6-week) of 102 dysregulated sterol lipids in each brain region were used for Pearson correlation calculation between two sterol lipids. In total, 77 sterol lipids and 223 coregulated pairs (220 positive and 3 negative correlation pairs) were retained after correlation filtering ($r \geq 0.85$ or $r \leq -0.85$). The coregulation network was constructed using Cytoscape (version 3.7.2). Nodes in the network are dysregulated sterol lipids, while colors indicate the log2 of fold-changes for each of the ten regions.

**Reporting summary**. Further information on research design is available in the Nature Research Reporting Summary linked to this article.

## Data availability

The raw data files generated in this study have been deposited in the National Omics Data Encyclopedia under accession code OEP002113. The converted data (MS1 peak table and MS/MS spectra files) generated in this study have been deposited in the MetaboLights under accession code MTBLS2457. The information on 97 sterol lipids in the standard ST library and 2068 sterol lipids in the extended ST library generated in this study is provided in Supplementary Data 3. The sterol identification and quantification results generated in this study are provided in Supplementary Data 6–8. The interactive mouse brain atlas of sterol lipids is provided on the Brain Sterol Atlas website (http://mousebrainatlas.zhulab.cn/). The simplified molecular-input line-entry specification (SMILES) of 97 sterol lipids were collected from LIPID MAPS Structure Database (https://www.lipidmaps.org/data/structure/download.php, accessed on December 16th, 2016) and PubChem (https://pubchem.ncbi.nlm.nih.gov/, accessed on December 26th, 2016) to describe the chemical structures. Source data are provided with this paper.

## Code availability

The source code of Sterol4DAnalyzer is provided in GitHub and Zenodo.

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

## Acknowledgements

The work is financially supported by the National Key R&D Program of China (2018YFA0800902), the National Natural Science Foundation of China (31971356 and 92057114), Shanghai Municipal Science and Technology Major Project (2019SHZDZX02), and the Chinese Academy of Sciences Major Facility-based Open Research Program. Z.J.Z. is supported by the Excellent Young Scholar Fund from the National Natural Science Foundation of China (22022411).

## Author contributions

Z.J.Z. and T.L. conceived the idea and designed the project. T.L., J.Q., and W.L. performed the sample preparation, data acquisition, and data processing. Y.Y. and Z.Z. contributed to software and database. T.L. and Y.C. performed the data analysis. X.Z. and K.H. contributed to mouse brain dissection. Z.J.Z., Y.C., and T.L. wrote the manuscript. Z.J.Z. supervised the project.

## Competing interests

The authors declare no competing interests.
