## [Peer Review File · Nature Communications]

REVIEWER COMMENTS

Reviewer #1 (Remarks to the Author):

This manuscript introduces a novel ion mobility-mass spectrometry approach facilitated by machine-learning for the profiling of a broadest range possible of sterol lipids in mouse brain tissue. For the methodological point of view, the authors have nicely demonstrated how taking advantage of four different dimensions of chemical information (i.e. accurate mass, CCS and RT values and MS/MS data) allowed for the characterization of more than 2000 sterol species, including numerous isomers. As a proof-of-principle they have showed the data acquired from plasma, brain and liver tissue analysis. For the biological point of view, several major issues need to be addressed.

Firstly, the authors should explain how exactly mice were sacrificed – by decapitation? It has been reported that the decapitation followed by immediate freezing causes a certain level of ischemia that could affect labile metabolites. This is even more true when the dissection is performed. This issue was extensively discussed in the following review:

<https://onlinelibrary.wiley.com/doi/full/10.1111/jnc.15293?campaign=wolearlyview>

This concern is even more relevant with respect to the fact that many sterols are subject to oxidation – this is also one of the reasons for which authors use BHT – “to avoid possible oxidation during sample preparation”. But what about sample collection? The dissection to 10 different regions has for sure taken at least 30 minutes? It is important that the authors discuss the potential limitations or at least the biological relevance of their findings – depending on the protocol used to harvest the brains and the impact that it has on sterol profile.

Another important issue is related to the investigation of the effect of “aging”. Aging is a process and cannot possibly be studied in independent samples (young vs. old mice). It can only be studied in a longitudinal way using paired samples. This should be carefully corrected throughout the manuscript – the authors have not studied the “changes during aging” or “related to aging”, they have only described the age-associated differences (rather than changes – the term which again implies a process and not a state) by comparing the sterol levels in young vs. old mice brain.

In general, all the findings remain descriptive and difficult to resume. The authors should consider of presenting the results in a web-based application where the users could easily interactively explore the levels of different steroids across different brain regions.

Reviewer #2 (Remarks to the Author):

General comments:

Non-targeted metabolomics is considered a powerful approach to reveal the mechanisms of diseases, aging, etc. However, metabolomics approaches are limited by the low rate of identified compounds. Therefore, analytical workflows such as the one presented by Li et al. in this manuscript contribute to generate more analytical information on molecules present in biological samples for their identification with greater confidence.

This manuscript presents a novel approach for the non-targeted analysis of sterol lipids in biological samples. It integrates liquid chromatography, ion mobility and mass spectrometry, with the combination of matching learning databases for the identification of a greater number of compounds. It worth mentioning the derivatization strategy followed to improve the separation of isomeric sterol lipids in the ion mobility dimension. As a novel approach, the composition (in terms of sterol lipids) of different parts of the brain is investigated. This study opens a new way of investigating brain composition and associate specific parts of it with the development of diseases, for example. In general, the manuscript is well and clearly presented. Although the results obtained are very

interesting from an analytical point of view, I am not completely convinced that the relationship between the variation of sterol lipids in brain and aging is well addressed (e.g. low number of samples investigated). Overall, there is a lack of critical discussion on the limitations of the proposed workflow, which I think will contribute to establishing the current limitations and required developments in ion mobility spectrometry, the collision cross section parameter, and databases based on matching learning approaches for biological analysis applying metabolomics.

In my opinion, this manuscript is suitable to be published in Nature Communications as it highlights the relevance of developments in analytical chemistry to adequately address current challenges found in other fields of research.

Specific comments:

- The use of the word 'sterolomics' does not appear to be appropriate as sterol lipids are part of the metabolome; therefore, they are investigated as part of metabolomics approaches, and furthermore the use of the word 'sterolomics' is not widely accepted. From a scientific point of view, it is not worthy to create new words for each study referring metabolomics approaches for specific families of compounds. Therefore, the term 'sterolomics' should be deleted or replaced.
- Page 4 – 'Introduction' section: The statement 'It enables to distinguish the isomeric compounds that commonly exist in biological samples', for ion mobility spectrometry, is too optimistic. In general, ion mobility spectrometry contributes to the separation of isomers, but a significant percentage of them show identical drift times or collision cross sections; therefore, they are not separated.
- Figure 1 – Replace '4D Sterolome identification' for '4D sterol lipids identification'
- Page 6-7 – 'Improved separation of sterol isomers with derivatization and IMS-MS'. It is clear that derivatization favors the detection of sodium adducts of derivatized compounds, whereas $[M+H-H_2O]^+$ or $[M+H-2H_2O]^+$ ions are detected for underivatized compounds. For many sterol lipids, the loss of water molecules implies the loss of isomerism. It is interesting to add a comment on this topic in the 'Results' section.
- Page 6-7 – 'Improved separation of sterol isomers with derivatization and IMS-MS'. It is necessary to include three-dimensional figures showing that isomer pairs were baseline resolved by LC-IM whereas poor resolution was observed when using IM separation and LC separation alone.
- The standard (ST) library (Supplementary Data 3) includes 98 compounds and not 97 sterol lipids. Verify that the number of compounds and percentages of them discussed on each step of the study is correct.
- Page 7 - Why were only 2068 sterol lipids retrieved from the LIPID MAPS Structure Database investigated, if it includes up to 2832 sterol lipids according to Figure 1?
- Page 7 – The percentage of compounds used as the training data set must be accurate. In total, 42 compounds were used as a training data set and represents 42.9% of the 98 compounds in the ST library (and not 40% of them).
- Page 7 – I am aware of the limitations of machine-learning algorithms for CCS prediction, and the authors should also discuss their limitations. A median relative error (MRE) of 1.75% for CCS prediction appears to be quite high considering that 2% is the current accepted threshold for CCS measurements against CCS databases. It appears that CCS databases created with these machine-learning algorithms are prone to generating a high percentage of false negatives and misidentifications. In Supplementary Data 4, it can be seen how CCS prediction leads to an error greater than 2% for 18 compounds of the 42 sterol lipids included in the training data set.
- How was the weight of RT, CCS and MS/MS spectral match scores selected? In my opinion, assigning the same weight to the CCS and MS/MS spectra (0.4) is a mistake as more information can be obtained from the MS/MS spectra than from CCS values.
- Figure 4.g – The predictive interval considered for the CCS vs m/z trend line in the chemical space appears to be too large. How was it established? More information is required on the confidence interval for the trend line.

- Figure 5.c – More information is required on what does each of the columns refer to.
- Page 12 – Since sterol lipid concentration measurements are not expected to be based on one measurement alone, the concentration values should include the related standard deviation.
- Page 12 – When discussing sterol lipid levels, it is recommended to avoid the terms ‘the least abundant’ or ‘the most abundant’ as other sterol lipids may not have been detected because they are at concentration levels below the limits of detection of the method. Furthermore, unidentified sterol lipids could be found at higher concentration levels greater than the putatively identified compounds.
- Page 13 – ‘Age-associated spatial diversity of sterol lipids in mouse brain’ section. The authors state that 102 sterol lipids were found to change significantly in at least one mouse brain region during aging. In my opinion, this evidence does not show that sterol lipid variations are caused by aging. If aging causes sterol lipid variations, this should be observed in all six 68-week-old mice examined.
- Discussion section – I do not agree with the statement ‘the relationship between secosterol-B and pathogenesis of AD has not been well studied, given its strong cytotoxic effects’. Although the relationship between this sterol lipid and the AD disease is unclear, it does not necessarily imply that studies have been limited due to the cytotoxicity of this compound.

Reviewer #3 (Remarks to the Author):

This is an excellent paper with a tremendous amount of information. I would have liked an attempt at uncovering the false discovery rate when using the extended sterol library.

The data and software are available and meet the requirements of the journal.

The paper is generally well written however a few typos are dispersed throughout. Here are a few:

Figure 1 panel B pyidine should read pyridine.

In brain tissue, six features were putatively annoated (annotated) as campesterol with MS1 match. The addition of RT match, CCS match and MS/MS spectraol (spectral) match filtered false positives and annotated the feature M528T849C269 as campesterol. This identification was validated using chemical (chemical) standard (Fig. 4d).

with MS1 macth (match) in brain tissue. The addtion of RT match, CCS match and MS/MS spectral match

The resutls demonstrated that all sterols falled (fell) within the sterol chemical space

Response to the reviewers:

The authors would like to thank the reviewers for the helpful comments. We believed that these comments have strengthened the manuscript considerably.

Reviewer #1:

Remark to the Author: *“This manuscript introduces a novel ion mobility-mass spectrometry approach facilitated by machine-learning for the profiling of a broadest range possible of sterol lipids in mouse brain tissue. For the methodological point of view, the authors have nicely demonstrated how taking advantage of four different dimensions of chemical information (i.e. accurate mass, CCS and RT values and MS/MS data) allowed for the characterization of more than 2000 sterol species, including numerous isomers.”*

Ans: We appreciate the positive comments from the reviewer.

Comment #1: *“As a proof-of-principle they have showed the data acquired from plasma, brain and liver tissue analysis. For the biological point of view, several major issues need to be addressed. Firstly, the authors should explain how exactly mice were sacrificed – by decapitation? It has been reported that the decapitation followed by immediate freezing causes a certain level of ischemia that could affect labile metabolites. This is even more true when the dissection is performed. This issue was extensively discussed in the following review: <https://onlinelibrary.wiley.com/doi/full/10.1111/jnc.15293?campaign=wolearlyview> This concern is even more relevant with respect to the fact that many sterols are subject to oxidation – this is also one of the reasons for which authors use BHT – “to avoid possible oxidation during sample preparation”. But what about sample collection? The dissection to 10 different regions has for sure taken at least 30 minutes? It is important that the authors discuss the potential limitations or at least the biological relevance of their findings – depending on the protocol used to harvest the brains and the impact that it has on sterol profile.”*

Ans: We appreciate the comments from the reviewer. We have taken the reviewer’s suggestion and added the details of mouse scarification and tissue dissection in our revised manuscript as following:

“Mice (C57BLJ6; female; 6- and 68- week, n=6 in each group) were first anesthetized with isofluorance vapor and sacrificed by decapitation. The whole brain was manually divided into different subregions by referring to the mouse brain atlas (Paxinos and Franklin, The Mouse Brain, 4th edition). The brain tissue was kept in ice-cold dissection buffer (212.7 mM sucrose, 5 mM KCl, 1.25 mM NaH₂PO₄, 10 mM MgCl₂, 0.5 mM CaCl₂, 26 mM NaHCO₃, and 10 mM dextrose, and bubbled with 95% O₂/5% CO₂ (pH 7.4)) during the whole process to prevent neural excitotoxicity and maintain viability. Tissue was directly frozen in liquid nitrogen and kept in -80 °C before further experiment. It took approximately 15 minutes to fully dissect one mouse brain immediately after euthanasia.”

We also agree with the reviewer that decapitation and dissection may cause the changes of labile metabolites during sample collection. To minimize post-mortem changes of sterol lipids, we followed the protocol of low temperature operation and rapid freezing of samples. In the revised manuscript, we have taken the reviewer's suggestion and added following discussion:

"It is worth noting that although we have carefully followed the protocol of performing the dissection experiments at a low temperature and rapidly freezing the samples during collection to minimize postmortem changes in sterol lipids that may occur in brain tissues, the possible oxidation of sterol lipids is unavoidable. This fact may have contributed to the sterol profiles reported in this work being partially dependent on the protocol used to harvest the brains and the impact on sterol profiles."

In addition, the suggested reference (Gerald A. Dienel, **J. Neurochem.**, 2021, <https://doi.org/10.1111/jnc.15293>) has also been added in the revised manuscript.

Comment #2: *"Another important issue is related to the investigation of the effect of "aging". Aging is a process and cannot possibly be studied in independent samples (young vs. old mice). It can only be studied in a longitudinal way using paired samples. This should be carefully corrected throughout the manuscript – the authors have not studied the "changes during aging" or "related to aging", they have only described the age-associated differences (rather than changes – the term which again implies a process and not a state) by comparing the sterol levels in young vs. old mice brain."*

Ans: Thanks for the reviewer's comment. In this study, it is not feasible to study aging longitudinally using the paired samples since the mice need to be sacrificed at each time point. We agree with the reviewer's comment and we have carefully revised throughout the manuscript using the suggested term, such as "age-associated differences".

Comment #3: *"In general, all the findings remain descriptive and difficult to resume. The authors should consider of presenting the results in a web-based application where the users could easily interactively explore the levels of different steroids across different brain regions."*

Ans: We thank reviewer's comment. In the revised manuscript, we have developed a web-based application using python tools to demonstrate the levels of sterol lipids across different brain regions (<http://mousebrainatlas.zhulab.cn/>). We have added the information in the revised manuscript.

Reviewer #2:

Remark to the Author: *“Non-targeted metabolomics is considered a powerful approach to reveal the mechanisms of diseases, aging, etc. However, metabolomics approaches are limited by the low rate of identified compounds. Therefore, analytical workflows such as the one presented by Li et al. in this manuscript contribute to generate more analytical information on molecules present in biological samples for their identification with greater confidence. This manuscript presents a novel approach for the non-targeted analysis of sterol lipids in biological samples. It integrates liquid chromatography, ion mobility and mass spectrometry, with the combination of matching learning databases for the identification of a greater number of compounds. It worth mentioning the derivatization strategy followed to improve the separation of isomeric sterol lipids in the ion mobility dimension. As a novel approach, the composition (in terms of sterol lipids) of different parts of the brain is investigated. This study opens a new way of investigating brain composition and associate specific parts of it with the development of diseases, for example. In general, the manuscript is well and clearly presented. Although the results obtained are very interesting from an analytical point of view, I am not completely convinced that the relationship between the variation of sterol lipids in brain and aging is well addressed (e.g. low number of samples investigated). Overall, there is a lack of critical discussion on the limitations of the proposed workflow, which I think will contribute to establishing the current limitations and required developments in ion mobility spectrometry, the collision cross section parameter, and databases based on matching learning approaches for biological analysis applying metabolomics. In my opinion, this manuscript is suitable to be published in Nature Communications as it highlights the relevance of developments in analytical chemistry to adequately address current challenges found in other fields of research.”*

Ans: We appreciate the positive comments from the reviewer towards publication. We have taken reviewer’s comments and added critical discussion on potential limitations of our work in the revised manuscript.

“Difficulties in sterol lipid identification have made the comprehensive characterization of sterol lipids in biological samples challenging. Our IM-MS based technology also has limitations in sterol identifications. First, due to the limitation of picolinyl derivatization, some important sterols such as cholesterol ester (CE) and steroids without hydroxyl groups were missing from the current analysis. Second, the limited number of available sterol standards impeded the accuracy of predicted CCS values in extended ST library. The low accuracy of predicted CCS values is prone to generate false sterol identifications. To minimize the effect, we employed the experimental CCS values of the standard ST library for identification of level 1 sterol lipids. However, for level 2 identifications, the false identifications are unavoidable since the predicted CCS values were used for matching... In the future, with the increase of available sterol standards, we could gradually expand the size of the training data set. We could also develop alternative machine-learning algorithms such as neural network algorithm to further improve the predicted accuracy of CCS values.”

Comment #1: *“The use of the word ‘sterolomics’ does not appear to be appropriate as sterol lipids are part of the metabolome; therefore, they are investigated as part of metabolomics approaches, and furthermore the use of the word ‘sterolomics’ is not widely accepted. From a scientific point of view, it is not worthy to create new words for each study referring metabolomics approaches for specific families of compounds. Therefore, the term ‘sterolomics’ should be deleted or replaced.”*

Ans: We appreciate the comment from the reviewer. “Sterolomics” is not a new word that we created. We share the same feeling with the reviewer that it is not necessary to create a new “omics” word for a specific class of compounds belonging to part of the metabolome (since the main research focus of our lab is untargeted metabolomics). In 2014, Roberg-Larsen *et al.*, first reported the word of “sterolomics” (*J. Lipid Res.*, 2014, <https://doi.org/10.1194/jlr.D048801>). Later, Griffiths *et al.*, who is a significant contributing scientist in sterol field, comprehensively defined the “sterolomics” in two recent publications (*Biochim. Biophys. Acta.*, 2017, <https://doi.org/10.1016/j.bbaliip.2017.03.001>; and *TrAC-Trends Anal. Chem.*, 2019, <https://doi.org/10.1016/j.trac.2018.10.016>). Recently, there are also other publications reporting the word of sterolomics (PMID: 29627611; PMID: 30366429; PMID: 31247920; PMID: 30736477).

In order to make sure that our publication is well accepted by scientists in sterol field, we therefore used the sterolomics in our manuscript. We share the same feeling with the reviewer. Thus, in the revised manuscript, we have replaced most “sterolomics” with “the analysis of sterol lipids”. Throughout the whole manuscript, we only keep three places using the word of “sterolomics”, including the title, the abstract, and the introduction.

Comment #2: *“Page 4 – ‘Introduction’ section: The statement ‘It enables to distinguish the isomeric compounds that commonly exist in biological samples’, for ion mobility spectrometry, is too optimistic. In general, ion mobility spectrometry contributes to the separation of isomers, but a significant percentage of them show identical drift times or collision cross sections; therefore, they are not separated.”*

Ans: Thanks for the reviewer’s comment. The sentence has been revised as “It contributes to the separation of isomeric compounds that commonly exist in biological samples” in the revised manuscript.

Comment #3: *“Figure 1 – Replace ‘4D Sterolome identification’ for ‘4D sterol lipids identification’”.*

Ans: Thanks for the reviewer’s comment. The words have been revised as “4D identification of sterol lipids” in Figure 1 in the revised manuscript.

Comment #4: *“Page 6-7 – ‘Improved separation of sterol isomers with derivatization and IMS-MS’. It is clear that derivatization favors the detection of sodium adducts of derivatized compounds, whereas $[M+H-H_2O]^+$ or $[M+H-2H_2O]^+$ ions are detected for underivatized compounds. For many sterol lipids, the loss of water molecules implies the loss of isomerism. It is interesting to add a comment on this topic in the ‘Results’ section.”*

Ans: Thanks for the reviewer’s excellent comment. Yes, we have noticed that, for underivatized compounds, the major adducts were $[M+H-H_2O]^+$ or $[M+H-2H_2O]^+$. We agree that the loss of water molecules for underivatized sterols possibly caused the loss of isomerism, which further generated small dereferences on CCS values between two isomers. We have taken the reviewer’s comment and added the following comment in the revised manuscript:

“It is worth noting that for underivatized sterol lipids of which the major adduct is $[M+H-H_2O]^+$ or $[M+H-2H_2O]^+$, the loss of water molecules may cause loss of isomerism”.

Comment #5: *“Page 6-7 – ‘Improved separation of sterol isomers with derivatization and IMS-MS’. It is necessary to include three-dimensional figures showing that isomer pairs were baseline resolved by LC-IM whereas poor resolution was observed when using IM separation and LC separation alone.”*

Ans: Thanks for the reviewer’s comment. We have added the three-dimensional plot to demonstrate the separation of isomers in the revised manuscript (**Figure 2c**, right panel).

Comment #6: *“The standard (ST) library (Supplementary Data 3) includes 98 compounds and not 97 sterol lipids. Verify that the number of compounds and percentages of them discussed on each step of the study is correct.”*

Ans: Thanks for the reviewer’s vigilant check and comment. We have carefully checked and found the duplicate records of one compound in the table. This has been corrected in the revised Supplementary Data 3. The final number of sterol standards in the library is confirmed as 97.

Comment #7: *“Page 7 - Why were only 2068 sterol lipids retrieved from the LIPID MAPS Structure Database investigated, if it includes up to 2832 sterol lipids according to Figure 1?”*

Ans: Thanks for the reviewer’s comment. To curate the extended ST library, we first retrieved a list of 2,832 sterol lipids from LIPID MAPS Structure Database (accessed on December 16th, 2016), and removed 667 sterol lipids with zero or multiple hydroxyl groups ($n>5$), wherein 2,165 sterol lipids were kept. Among the 2,165 sterol lipids, 97 of them belong to the standard ST library, while the rest 2,068 sterol lipids were included in the extended library. We described this process in the “**Methods-Curation of the extended ST library**” (page 20-21).

Comment #8: “Page 7 – The percentage of compounds used as the training data set must be accurate. In total, 42 compounds were used as a training data set and represents 42.9% of the 98 compounds in the ST library (and not 40% of them).”

Ans: Thanks for the reviewer’s comment. The sentence has been revised as “*In brief, 57% of sterols in the standard ST library were used as a training data set, and the remaining 43% served as an external validation data set*”.

Comment #9: “Page 7 – I am aware of the limitations of machine-learning algorithms for CCS prediction, and the authors should also discuss their limitations. A median relative error (MRE) of 1.75% for CCS prediction appears to be quite high considering that 2% is the current accepted threshold for CCS measurements against CCS databases. It appears that CCS databases created with these machine-learning algorithms are prone to generating a high percentage of false negatives and misidentifications. In Supplementary Data 4, it can be seen how CCS prediction leads to an error greater than 2% for 18 compounds of the 42 sterol lipids included in the training data set.”

Ans: Thanks for the reviewer’s comments. Due to the limited number of sterol lipids in the training data set, the predicted error for sterol CCS values is relatively high (MRE=1.75%). We agreed that the low accuracy of predicted CCS values is prone to generate false negatives in sterol identifications. To minimize the effect, we employed the experimental CCS values of the standard ST library for identification of level 1 sterol lipids. However, for level 2 identifications, the false negatives are unavoidable since the predicted CCS values were used for matching. In the future, with the increase of available sterol standards, we could gradually expand the size of the training data set. In addition, we could also develop alternative machine-learning algorithms such as neural network algorithm to improve CCS accuracy. Our group are working on this and aim to develop the second generation of AllCCS2 (<http://allccs.zhulab.cn/>). We have added these discussions in the revised manuscript.

Comment #10. “How was the weight of RT, CCS and MS/MS spectral match scores selected? In my opinion, assigning the same weight to the CCS and MS/MS spectra (0.4) is a mistake as more information can be obtained from the MS/MS spectra than from CCS values.”

Ans: Thanks for the reviewer’s comment. The weighting parameters were originally developed in our previous publications (*Bioinformatics*, 2019, <https://doi.org/10.1093/bioinformatics/bty661>; *Nature Commun.*, 2020, <https://doi.org/10.1038/s41467-020-18171-8>; *Anal. Chim. Acta.*, 2020, <https://doi.org/10.1016/j.aca.2020.08.048>). For lipidomics analysis, since the MS/MS spectra in the library were *in-silico* predicted, which were not as accurate as experimental MS/MS library, the weighting factor of MS/MS spectral match was set as 0.4. In this study, the fragment ions for

derivatized sterol lipids were also predicted, we therefore used the same parameter setting. Of note, the weighting parameters were not fixed and can be adjusted by users in the software. Users are encouraged to increase the weight of MS/MS spectral matching if an experimental MS/MS library is used for matching (similar to most metabolomics studies). We have added the related discussion in the revised manuscript.

Comment #11. “Figure 4.g – The predictive interval considered for the CCS vs m/z trend line in the chemical space appears to be too large. How was it established? More information is required on the confidence interval for the trend line.”

Ans: Thanks for the reviewer’s comment. We fitted a trend line for sterol lipids using a power function with the standard ST library ($y=22.2x^{0.38}$, $R=0.7149$; $n=97$; Fig. 4g), and constructed the sterol chemical space with the 99% predictive interval. The construction of trend line using the power function is common in the IM-MS based metabolomics and lipidomics (*Anal. Chem.*, 2014, <https://pubs.acs.org/doi/10.1021/ac4038448>; *Chem. Sci.*, 2017, <https://doi.org/10.1039/c7sc03464d>; *Anal. Chem.*, 2017, <https://pubs.acs.org/doi/10.1021/acs.analchem.7b02625>; *Chem. Sci.*, 2018, <https://doi.org/10.1039/c8sc04396e>; *Anal. Chem.*, 2018, <https://pubs.acs.org/doi/10.1021/acs.analchem.7b05117>). We have added the detailed method information in the revised manuscript in the “**Methods-Construction of trend line of the standard ST library**” (Page 23) as the following:

“Construction of trend line of the standard ST library. The trend line was fitted using the nonlinear least square function “nls” in R (version 4.0.2). The data was retrieved from the standard ST library ($n=97$). The trend line was described as a power function $y=a*x^b$, where, the x is the precursor m/z value of sterol lipid, and the y is the precursor CCS value of sterol lipid. The starting point was set as ($a=1$, $b= 0.05$). Finally, the trend line of sterol lipids was fitted as ($y=22.2x^{0.38}$; $R=0.7149$; see Fig. 4g). Second, the chemical space was constructed by the trend line with 99% predictive interval (PI). The 0.99 PI was calculated by the following equation:

$$\Delta y(0.99PI) = Z \cdot S_{y,x} \cdot \left(1 + \frac{1}{n} + \frac{(x - \bar{x})^2}{SS_x}\right)^{1/2}$$

$Z = 2.576$, which is the standard deviation z-score based on 99% interval percentage;

$n = 97$, which is the number of data points (i.e., sterol lipid numbers);

$S_{y,x} = 17.94513$, which is the standard error of the x and y data inputs;

$SS_x = 98.71926$, which is the sum of the squared deviations from the mean of x inputs.”

Comment #12. “Figure 5.c – More information is required on what does each of the columns refer to.”

Ans: Thanks for the reviewer’s comment. In Figure 5c, each column represents the number of sterol lipids. In the lower part, the black solid dots represent the distributions of sterol lipids in different brain regions. Taken together, the first column indicated that there are 51 sterol lipids distributed in all of

ten brain regions. As a comparison, the second column indicated that there are 14 sterol lipids uniquely distributed in pons. This type of illustration in Figure 5c is named as “UpSet Plot”, and the detailed description can be found at website (<https://jku-vds-lab.at/tools/upset/>). We have added more detailed explanations in the revised manuscript.

Comment #13. *“Page 12 – Since sterol lipid concentration measurements are not expected to be based on one measurement alone, the concentration values should include the related standard deviation.”*

Ans: Thanks for the reviewer’s comment. The standard deviations of concentrations of sterol lipids have been added into the revised manuscript and supplementary information.

Comment #14. *“Page 12 – When discussing sterol lipid levels, it is recommended to avoid the terms ‘the least abundant’ or ‘the most abundant’ as other sterol lipids may not have been detected because they are at concentration levels below the limits of detection of the method. Furthermore, unidentified sterol lipids could be found at higher concentration levels greater than the putatively identified compounds.”*

Ans: Thanks for the reviewer’s comment. We have revised these sentences in the revised manuscript. The descriptions have been revised as *“the least abundant one of sterol lipids detected in this study”* or *“the most abundant one of sterol lipids detected in this study”*.

Comment #15. *“Page 13 – ‘Age-associated spatial diversity of sterol lipids in mouse brain’ section. The authors state that 102 sterol lipids were found to change significantly in at least one mouse brain region during aging. In my opinion, this evidence does not show that sterol lipid variations are caused by aging. If aging causes sterol lipid variations, this should be observed in all six 68-week-old mice examined.”*

Ans: Thanks for the reviewer’s comment. We are sorry that our description may have caused the misunderstanding. The sentence has been revised as *“We found that 102 sterol lipids identified in six mice samples have significant age-associated differences in at least one brain region”*.

Comment #16. *“Discussion section – I do not agree with the statement ‘the relationship between secosterol-B and pathogenesis of AD has not been well studied, given its strong cytotoxic effects’. Although the relationship between this sterol lipid and the AD disease is unclear, it does not necessarily imply that studies have been limited due to the cytotoxicity of this compound.”*

Ans: Thanks for the reviewer’s comment. The sentence *“given its strong cytotoxic effects”* has been removed in main text.

Reviewer #3:

Remark to the Author: *“This is an excellent paper with a tremendous amount of information. I would have liked an attempt at uncovering the false discovery rate when using the extended sterol library. The data and software are available and meet the requirements of the journal.”*

Ans: We appreciate the comments from the reviewer. To determine the false discovery rate using the extended ST library, we manually checked the results in olfactory bulb since this brain region has the most identified sterol lipids. After manual verification of each annotation with or without CCS match using the extended ST library, we calculated the FDR as the number of false positives (FP) divided by the sum of true positives (TP) and false positives (FP). The results showed that our four-dimensional matching method for identification of sterol lipids has a FDR value of 1.4% when the CCS match tolerance was set as 3% (the parameter used in our study). In the revised manuscript, we have added the false discovery rate (FDR) in the Discussion section and Supplementary Fig.9 as the following:

*“To examine the identification accuracy using the extended ST library, we estimated the false discovery rate (FDR) of sterol lipids in the olfactory bulb sample which is the sterol-richest brain region. With the addition of CCS matching for sterol lipid identification, the FDR decreased dramatically (**Supplementary Fig. 9**). The estimated FDR value in the data set was 1.4% when the CCS match tolerance was set as 3%. Thus, the results showed that our four-dimensional method enabled the characterization of sterol lipids in biological samples with high annotation accuracy.”*

Comment #1. *“The paper is generally well written however a few typos are dispersed throughout. Here are a few:*

Figure 1 panel B pyidine should read pyridine. In brain tissue, six features were putatively annoated (annotated) as campesterol with MS1 match. The addition of RT match, CCS match and MS/MS spectraol (spectral) match filtered false positives and annotated the feature M528T849C269 as campesterol. This identification was validated using chemcial (chemical) standard (Fig. 4d). with MS1 macth (match) in brain tissue. The addtion of RT match, CCS match and MS/MS spectral match. The resutls demonstrated that all sterols falled (fell) within the sterol chemical space.”

Ans: Thanks for the reviewer's comments. We have carefully checked and corrected the spelling in the revised manuscript and supplementary information. Following the suggestion, we have used the language editing service from the **Nature Research Editing Service** to improve the language quality of our manuscript. Thanks again for the reviewer's comments.

REVIEWERS' COMMENTS

Reviewer #1 (Remarks to the Author):

The authors have addressed all my comments and modified the main manuscript accordingly. They have also developed a web-based application that allows for the interactive data exploration. With respect to the effort that was put into the implementation of this application, the authors need to consider to make this application more useful. The point is that the user can explore how the levels of selected sterol species vary across different brain regions. This should be reflected on the brain section scheme across different regions - as relative content. For example, if we chose "cholesterol" we should be able to visualize that its' concentration is much higher in pons than in cerebellum (in the same way as in Figure 5f- h) The legend with a color code (or similar) should be added, reflecting the concentration levels. Importantly, the units in which the sterol concentrations were reported and how the reported concentrations were normalized must be added. Was the normalization done to tissue weight or to the protein content? This should be indicated, as for example 100 nmol / g of proteins. For the species that were not quantified, the authors should indicate the ion counts as arbitrary units. It is also advised to add the graphics that would illustrate relative differences among different regions - such a heat map, or violin plots for example (on which we could also see the data distribution). Finally, we should be able to explore the variation with age. With respect to small amplitude of differences (or fold changes), my advice is to use the "% of difference" instead.

Reviewer #2 (Remarks to the Author):

The authors have addressed all the reviewer's comments appropriately and made modifications to the manuscript in accordance with them. The manuscript can be published in Nature Communications in its current form.

Reviewer #1:

Remark to the Author: "With respect to the effort that was put into the implementation of this application, the authors need to consider to make this application more useful."

Ans: We appreciate the positive comments from the reviewer. We have made the suggested modifications to the website (<http://mousebrainatlas.zhulab.cn/>).

Comment #1: "The point is that the user can explore how the levels of selected sterol species vary across different brain regions. This should be reflected on the brain section scheme across different regions - as relative content. For example, if we chose "cholesterol" we should be able to visualize that its' concentration is much higher in pons than in cerebellum (in the same way as in Figure 5f- h)"

Ans: Thanks for the reviewer's comment. We added the bar plot to show the concentrations of the selected sterol species across different brain regions. The bar plot and brain regional heat map can reflect the different concentrations of the selected sterol.

Comment #2: "The legend with a color code (or similar) should be added, reflecting the concentration levels."

Ans: We thank reviewer's comment. The color code has been added to the graph to reflect the concentrations.

Comment #3: “Importantly, the units in which the sterol concentrations were reported and how the reported concentrations were normalized must be added. Was the normalization done to tissue weight or to the protein content? This should be indicated, as for example 100 nmol / g of proteins.”

Ans: We thank reviewer’s comment. We have added the unit (ng/mg) for the measured concentrations. The concentrations of sterol lipids were normalized to tissue weight, referring to ng of sterol per mg of tissues. The description of concentration unit was added to the website.

Comment #4: “For the species that were not quantified, the authors should indicate the ion counts as arbitrary units.”

Ans: Thanks for the reviewer’s comment. All identified sterol lipids in our paper were quantified.

Comment #5: It is also advised to add the graphics that would illustrate relative differences among different regions - such a heat map, or violin plots for example (on which we could also see the data distribution).”

Ans: We thank reviewer’s comment. We provided brain regional heat maps to illustrate relative differences among different regions.

Comment #6: “Finally, we should be able to explore the variation with age. With respect to small amplitude of differences (or fold changes), my advice is to use the “% of difference” instead.”

Ans: We thank reviewer’s comment. We provided $\log_2(\text{FC})$ of identified sterol lipids in bar plot to show the variation with age in 10 brain regions. The heat maps were also provided to reflect the relative differences among different regions.

Reviewer #2:

Remark to the Author: “The authors have addressed all the reviewer’s comments appropriately and made modifications to the manuscript in accordance with them. The manuscript can be published in Nature Communications in its current form.”

Ans: We thank the positive comments from the reviewer.